



# Measurement report: Properties of aerosol and gases in the vertical profile during the LAPSE-RATE campaign

David Brus[1], Jani Gustafsson[1], Ville Vakkari[1,2], Osku Kemppinen[3,†], Gijs de Boer[4,5], and Anne Hirsikko[1]

[1]Finnish Meteorological Institute, Erik Palménin aukio 1, P.O. Box 503, FIN-00100 Helsinki, Finland

[2]Atmospheric Chemistry Research Group, Chemical Resource Beneficiation, North-West University, South Africa

[3]Kansas State University, Department of Physics, 1228 N. 17th St., 66506, Manhattan, Kansas, USA

[4]University of Colorado, Cooperative Institute for Research in Environmental Sciences, 216 UCB, 80309, Boulder, Colorado, USA

[5]National Oceanic and Atmospheric Administration, Physical Sciences Division, 325 Broadway, 80305, Boulder, Colorado, USA

[†]Currently at University of Maryland, Earth Systems Science Interdisciplinary Center, 5825 University Research Ct suite 4001, College Park, MD 20740

**Correspondence:** David Brus (david.brus@fmi.fi)

**Abstract.** Unmanned Aerial Systems (UASs) are increasingly being used as observation platforms for atmospheric applications. The Lower Atmospheric Process Studies at Elevation - a Remotely-piloted Aircraft Team Experiment (LAPSE-RATE) in Alamosa, CO, USA during July 14th - 20th, 2018 investigated and validated different UASs, measurement sensors and setup configurations. Flight teams from the Finnish Meteorological Institute (FMI) and the Kansas State University (KSU) partici-

pated in LAPSE-RATE to measure and investigate properties of aerosol particles and gases in the lower atmosphere. During the experiment, the performance of different UAS configurations were investigated and confirmed to operate reliably resulting in a scientifically sound observational dataset. As an example, concentration of aerosols - including two new particle formation events, $CO_2$ and water vapor, and meteorological parameters in the atmospheric vertical profile were measured during the short experiment. Such observations characterizing atmospheric phenomena of this specific environment would have not been

possible in any other way, and thus, demonstrate power of UASs as new, promising tools in atmospheric and environmental research.

## 1   Introduction

Air pollution, including atmospheric aerosol particles and gases are released at the ground level from various sources. The

dispersion, transport and removal from the atmosphere of these particles and gases depends very much on wet deposition and atmospheric dynamical properties such as turbulent mixing (Tunved et al., 2013). The mixing height of the atmospheric boundary layer and wind speed and direction are influenced by a number of factors, including the amount of solar and surface



stored energy and terrain inhomogeneity (Carbone et al., 2010). The atmospheric boundary layer can be investigated through
various means, including balloon-borne soundings, tethersondes, dropsondes and hot-air balloons (e.g. Laakso et al., 2007;
Greenberg et al., 2009; Nygård et al., 2017), towers (e.g. Heintzenberg et al., 2011; Andreae et al., 2015), Unmanned Aerial
Systems (UASs) (e.g. Ramanathan et al., 2007; Jonassen et al., 2015; Kral et al., 2018; Barbieri et al., 2019; de Boer et
al., 2020a), remote-sensors, including ceilometers, sodars, Doppler lidars and radar techniques (e.g. O'Connor et al., 2010;
Schween et al., 2014; Hirsikko et al., 2014; Vakkari et al., 2015) and conventional research aircrafts (Hermann et al., 2003;
Twohy et al., 2002; Benson et al., 2008).

The Lower Atmospheric Process Studies at Elevation - a Remotely-piloted Aircraft Team Experiment (LAPSE-RATE; de
Boer et al., 2020a) campaign broadly investigated the influence of continental, inhomogeneous terrain on lower atmospheric
properties, land-air interactions, and the vertical distribution of aerosol particle and gas properties. LAPSE-RATE took place
in the greater San Luis Valley (SLV) of Colorado during July 14[th] - 20[th], 2018. The experiment was organized in conjunction
with the 6[th] Conference of the International Society for Atmospheric Research using Remotely-piloted Aircraft (ISARRA) and
included over 100 participants from a number of universities, governmental and industrial teams. These participants supported
the coordinated deployment of 35 different UAS, completing over 1300 flights and 250 flight hours in less than a week.
Distributed research flight plans were made and carried out to observe several atmospheric phenomena, including evolution
of the atmospheric boundary layer during morning hours, the diurnal cycle of valley flows, convective initiation, and also
concentration of gases and properties of aerosol particles.

LAPSE-RATE flights were conducted under both Federal Aviation Administration (FAA) Certificates of Authorization
(COAs) and FAA Part 107, with the COAs supporting flights up to altitudes of 914 m (3000 feet) above the ground level.
In addition to the aerial assets, a variety of ground-based observational assets were deployed including the University of Col-
orado's Mobile UAS Research Collaboratory (MURC), the University of Oklahoma Collaborative Lower Atmospheric Mobile
Profiling System (CLAMPS; Klein et al., 2015), two University of Colorado Doppler lidar systems, numerous radiosondes,
and mobile surface instrumentation associated with vehicles and small towers deployed by a variety of institutes. For additional
information on LAPSE-RATE, please see de Boer et al. (2020a, b, c) and Bell et al. (2020).

A flight team including two people from the Finnish Meteorological Institute (FMI) and one person from Kansas State
University (KSU) participated in LAPSE-RATE. This publication summarizes the measurement setups and observations made
by the FMI and KSU teams during the campaign. Section 2 provides an overview of the deployed rotorcraft and measurement
modules, as well as an introduction of the validation techniques applied to the deployed sensors. A summary of the collected
meteorological, aerosol and gas observations demonstrates the benefits of leveraging UAS for making measurements of these
quantities (section 3). Finally, a concluding section outlines overall successes of the aerosol and gas payloads deployed by FMI
and KSU during LAPSE-RATE. As a summary, the deployment of UAS-borne sensors provided new perspectives which could
have not been collected with other profiling methods (e.g. ground-based remote-sensing instrumentation).





## 2   Materials and Methods

During LAPSE-RATE, the FMI-KSU team was stationed at a sampling site along the County Road 53, approximately 15 km
north from Leach Airport (37°54'32.94"N, 106°2'6.83"W, see Fig. 1). The location was generally very quiet, surrounded by
farmland on all sides, and only sporadically disturbed by passing local farm trucks. The elevation of the operations site was
2291 m (7516 feet) above the mean sea level (MSL). The FMI team conducted flights under a FAA COA with a maximum
allowed altitude of 914 m (3000 feet) above the ground level (AGL). The KSU team conducted flights under FAA Part 107
with a maximum altitude of 121 meters (400 feet) AGL.

### 2.1   Technical description of rotorcraft

The FMI team deployed two rotorcraft (FMI-PRKL1 and FMI-PRKL2) during LAPSE-RATE. Both of these aircraft shared the
same configuration and were custom-built around the Tarot X6 hexacopter platform. The maximum endurance of these rotor-
craft was about 15 minutes, using brand new 22.2 V, 16000 mAh rechargeable lithium polymer (LiPo) batteries. A maximum
gross takeoff weight of the rotorcraft was close to 11 kg.

   Flight missions were carried out in a manual (stabilized) or loiter (GPS position fix) mode, and autonomous flight missions
were not carried out during the campaign. The flights were conducted using a 3DR Pixhawk PX4 flight controller running the
Ardupilot software. The rotorcraft airframe was 960 mm (rotor-to-rotor) in diameter. Both rotorcraft used the same propulsion
system consisting of 340 kV brushless motors, 40A Electronic Speed Controllers and 18 inch (5.5 inch pitch) carbon fiber
propellers. Such a setup supported lifting approximately 2 kg of active payload (i.e. scientific instrumentation). The rotorcraft
also utilized a first-person viewer (FPV) video link, allowing real-time video broadcast from the aircraft to be viewed on a
ground station monitor. The video link provided visual confirmation of the UAV performance together with telemetry infor-
mation overlaid via an On Screen Display (OSD). UAV flight performance was recorded with a high resolution Digital Video
Recorder (DVR) and stored for post-flight analysis. Finally, to avoid any high energy impacts and to protect the expensive
scientific instrumentation on board in case of a multi-level safety system failure, we employed a parachute system using a 60"
parachute rated to 7 m.s$^{-1}$ descent speed for a 12 kg UAV. The parachute system had an independent power system and was
configured for both manual and automatic launch.

   The KSU rotorcraft was a DJI Matrice 600 Pro without any modifications, aside from payload attachments. The rotorcraft
was controlled with the Matrice 600 Pro remote controller. Both DJI TB47S (6×4500 mAh, 22.2 V) and DJI TB48S (6×5700
mAh, 22.8 V) rechargeable LiPo batteries were used, alternating between battery types between flights. The maximum gross
take-off weight recommended by the manufacturer was 15.5 kg, which would have resulted in a maximum payload capacity
of roughly 5.5 kg. However, such a take-off weight would likely have been challenging given the high elevation of San Luis
Valley. During all flights the rotorcraft was additionally equipped with a DJI Zenmuse X3 camera, and an additional GoPro
HERO 2018 on some flights to provide alternative visual perspectives.



## 2.2    Description of payload modules

Both FMI rotorcraft carried custom-built payload modules consisting of a carbon fiber frame made out of tubes (O.D.=12 mm). A carbon fiber plate (30×20 cm and 2 mm thick), on which scientific instruments were attached, was centrally-mounted on the module. The 33×30×20 cm (H×W×D) frame was attached between the rotorcraft's retractable landing gear on the rails supporting the battery plate (Fig. 2 and 3). These modules were easily detached from the rotorcraft frame, allowing for swapping between sensor modules to meet the requirements of a given flight. The first FMI rotorcraft (FMI-PRKL1) was equipped with a FMI-developed particle measurement module. This module consisted of two condensational particle counters (CPC, model 3007, TSI Corp., total count in range from 0.01 to >1.0 μm), a factory-calibrated optical particle counter (OPC, model N2, Alphasense, 16 bins size resolved range from 0.38 to 17 μm) and a basic meteorological sensor (P, T and RH). The CPCs were calibrated to different cut-off diameters (7 and 14 nm, respectively, c.f. Altstädter et al., 2015, 2018). Such a configuration enables observation of freshly nucleated particles in a diameter range between 7 to 14 nm. The voltage applied to the Thermal Electric Device (TED) of a CPC corresponds directly to the temperature difference between the saturator and condenser. This temperature difference determines how fast particles grow to CPC detectable sizes (i.e. the cut-off diameter). Therefore, limitations to instrumental stability and sensitivity to changing saturator-condenser temperature differences, limited the ability to observe the full nucleation mode range up to 20 nm. TED values of 2000 and 1000 mV were used for CPC1 and CPC2, respectively. In addition to the CPCs and OPC, FMI-PRKL1 included an Arduino Bosh BME280 sensor, located below the modular platform, to measure pressure, temperature and humidity. The platform was covered from all sides except the bottom, with polylactide (PLA) foam to protect sensors from solar radiation and keep the particle module thermally stable. The BME280 sensor was shielded from solar radiation, but not forcefully aspirated. The BME280 sensor has a manufacturer-stated response time and accuracy of 6 ms, ±1 hPa for pressure, 1s, ±0.5 °C for temperature, and 1s, ±3% RH for relative humidity.

The second FMI rotorcraft (FMI-PRKL2) was equipped with a second FMI-developed module, carrying instrumentation to measure gases and basic meteorological parameters (P, T and RH). The gas module consisted of a flow-through $CO_2$ concentration sensor (Carbocap model GMP343, Vaisala Inc.), a $CO_2$ and water vapor analyzer (model Li-840A, Li-Cor Envir.) and a sensor to measure concentrations of CO, $NO_2$, $SO_2$, and $O_3$ (model AQT400, Vaisala Inc.). An Arduino BME280 sensor measured pressure, temperature and humidity. The BME280 sensor was mounted identically to the respective sensor on the FMI-PRKL1. Both BME280 sensors showed a difference of about +2 hPa in pressure, +2 °C in temperature, and -12% in RH, when compared to measurements from the MURC platform during LAPSE-RATE inter-comparison flights (for details please see Barbieri et al. (2019). Both $CO_2$ sensors were forcefully-aspirated using micro-blowers configured as air pumps (Murata, model MZB1001T02) and leveraging custom 3D-printed housings connected to the sensor exhaust. A Gelman filter in the sample airstream in front of both $CO_2$ sensors prevented contamination of the optical path. Micro-blowers were supplied with 12V DC resulting in a constant flow rate of 0.6 L.min$^{-1}$ to continually flush measured air through the sample cells. The GMP343 and LI-840A are non-dispersive infrared (NDIR) gas analyzers based upon a single path, dual wavelength infrared detection system. The GMP343 has a manufacturer stated accuracy of ±3 ppm ±1% of the measured reading and the LI-840A has an accuracy better than 1.5% of the measured reading for both $CO_2$ and water vapor.





Both $CO_2$ sensors were laboratory-calibrated before and after the LAPSE-RATE campaign and showed no drift in calibration. The sensors were calibrated against standard carbon dioxide gases (traceable to WMO $CO_2$ scale X2007 at the FMI) at several concentrations (zero gas and 436 ppm before the campaign, and 370, 405.4 and 440.2 ppm after the campaign). Also, both sensors were tested in the lab against a calibrated high precision gas concentration analyzer (model G2401, Picarro, Inc.) at ambient $CO_2$ concentration. The GMP343 data were biased on average -3.4 (±1.3) and LI-840A 0.3 (±1.1) ppm. Data from

the GMP343 sensor were post-processed to account for pressure, temperature, RH (obtained from BME280 sensor) and oxygen by using the proprietary compensation algorithm of the probe obtained from Vaisala. For the Licor LI-840A, the built-in option for internal pressure, temperature and water vapor compensation of $CO_2$ concentration was used. Licor LI-840A water vapor was calibrated in dew point measurements against DewMaster Chilled Mirror Hygrometer (Edgetech Instruments Inc.).

    The AQT400 (Vaisala Inc.) gas sensor is based on proprietary advanced algorithms and electrochemical technology. The

sampled gas causes a reduction or oxidation reaction in an electrochemical cell, thereby creating a weak electric current that is directly dependent on the volume of the measured gas. The gas content is subsequently calculated using an advanced compensation algorithm. The detection limits of the Vaisala AQT400 are 5 ppb for $NO_2$, $O_3$ and $SO_2$ and 10 ppb for CO, and the in-field accuracy is ±25, 60, 200 and 50 ppb for $NO_2$, $O_3$, CO and $SO_2$ respectively.

    A third FMI-developed sensor module was operated on the ground, consisting of a condensational particle counter (CPC,

model 3007, TSI Corp.), an optical particle counter (OPC, model N2, Alphasesne) and a TriSonica Mini Weather Station (Applied Technologies, Inc.). The manufacturer stated accuracies for pressure, temperature and RH are ±3 hPa, ±0.5 °C, and ±3% RH, respectively. The surface sensor module was covered from all sides with PLA foam to protect sensors from solar radiation and keep the particle module thermally stable. The module was placed on the roof top of a car at about 2 m from the ground, the TriSonica Mini WS was mounted on a 45 cm long carbon fiber tube on top of the surface sensor module, i.e. about

2.75 m from the ground and about 75 cm from the car roof.

    All FMI-PRKL1 particle module and meteorological sensors were logged at a rate of 1 Hz, except the OPC-N2 that was logged at rate of 0.5 Hz. The sensors included in the FMI-PRKL2 gas module were logged at different rates, with the meteorological variables, $CO_2$ and water vapor analyzer (LI-840A) logged at 1 Hz, the flow-through $CO_2$ concentration sensor (Carbocap, GMP343) logged at 0.5 Hz, and the gas sensor (AQT400, Vaisala Inc.) logged once per minute. The surface sensor

module data were logged at 1 Hz, except the OPC-N2 that was logged at rate of 0.5 Hz. Data from both rotorcraft and the surface sensor module were logged as ASCII comma separated files to embedded Raspberry Pi 3+ minicomputers using simple Python scripts. This information was aligned with position and orientation information from the autopilot in post-processing, using cross-correlation techniques to align the signals in time.

    The KSU Matrice 600 Pro was equipped with an optical particle counter (POPS, Handix Scientific LLC) with 16 size bins

covering a range from 0.132 to 3.648 μm and measuring at 1 Hz. During LAPSE-RATE the POPS was used both with a horizontally oriented naked inlet with inner diameter of 1.7 mm (0.069 inches), and a vertically oriented tube inlet of approximately 45 cm (18 inches) long and with an inner diameter of 3.175 mm (0.125 inches). POPS included custom electronics, and its data were logged to a microSD card. POPS was attached to the top surface of the rotorcraft body with Velcro tape. It was not





shielded from direct sunlight during the flights, but was kept in shade while on the ground. A duplicate POPS instrument was

operated as a ground reference and was located approximately 1.8 m above the ground level.

## 2.3    Auxiliary data

Air mass history was investigated using the Lagrangian particle dispersion model Flexpart, version 9.02 (Seibert and Frank, 2004; Stohl et al., 2005). European Centre for Medium-Range Weather Forecasts (ECMWF) operational forecasts were used as meteorological input to the Flexpart back trajectories. The spatial resolution of ECMWF forecast is 9 km horizontally and

variable in the vertical (137 model levels) and we obtained the model fields at 0.1 degree latitude-longitude resolution for an area of 10°N - 75°N and 175°W - 70°W. The temporal resolution of this ECMWF product is 1 hour. We obtained potential emission sensitivity fields from backwards simulations with Flexpart to estimate air mass history (Seibert and Frank, 2004). The simulations were run 96 hours backwards and output was saved at 30 min time resolution. We used also mixing height of the atmospheric boundary layer (mixing layer height = MLH) from the ECMWF operational forecast calculated as the mean

of four nearest model grid points to the measurement location.

## 3    Results and Discussion

Altogether, the FMI team completed 38 vertical profile flights. Of these, 14 flights were completed with the particle module and 24 flights with the gas module. The maximum achieved height was 3201 m MSL (i.e. 893 m AGL). Vertical profiles were performed by alternating flights between the particle and gas modules, with approximately 30 minutes between flights. The

ground module logged continuously during the whole flight operation period. The first six flights conducted on July 15[th] were test flights designed to evaluate and fix several issues that arose due to transport of equipment between Finland and Colorado. The KSU team completed in total 33 flights with their payload, including 40 individual vertical profiles. It should be noted that some of these profiles were redundant, made within a few minutes of each other, and repeated in the exact same location as another profile. The redundant flights were completed to test consistency of the instrument. All profiles were measured up to

121 meters AGL. During the first three days (July 15[th]-18[th]), the FMI-KSU team flight missions focused on vertical profiling of aerosol particle and gas properties. During the last sampling day (July 19[th]), the FMI-KSU team joined the other LAPSE-RATE participants in order to conduct flights evaluating cold-air drainage from local valleys during the morning hours.

### 3.1    Ground observations

### 3.1.1    Meteorological variables

Weather forecasting and modeling support for the LAPSE-RATE campaign was provided by the National Weather Service forecast office in Pueblo, CO and the National Center for Atmospheric Research (NCAR). In general the San Luis Valley is dry during summer months, though convections occur frequently over the surrounding mountains. A convection results in an afternoon thunderstorm which, in favorable wind and moisture conditions, can occasionally precipitate over the valley. The



detailed description of day-to-day meteorological situation in SLV during LAPSE-RATE campaign was given in de Boer et al.
(2020a).

The surface sensor module measured continuously meteorological variables (temperature, relative humidity, pressure, wind speed and direction) and aerosol concentration (CPC and OPC-N2). We compared observed meteorological variables to those obtained with MURC. MURC was equipped with a 15-meter high expandable mast mounted with several meteorological sensors including a Gill MetPak Pro Base Station that provided barometric pressure, temperature, and humidity; a Gill 3D sonic
anemometer for 3D wind measurements and an R.M. Young Wind Monitor anemometer that provided a redundant horizontal wind measurement (see de Boer et al., 2020c). During the majority of the campaign, MURC was positioned at the Leach Airport, about 15 km south from our location (Fig. 1). The MURC tower was the only reference measurement point in the vicinity of our location. Figures 4 A) and B) present diurnal temperature and relative humidity measured with TriSonica mini weather station mounted on FMI surface module together with MURC data for the same time span. Comparison of observed
temperatures suggests that just before midday, when ambient air temperature exceeds 20 °C and is likely homogeneously distributed over a large area, differences between values were within accuracy of sensors. During morning hours localized gradients in ambient conditions were more pronounced, resulting in significant differences (up to 4 °C) between observations at the two locations. Observations for pressure follow similar diurnal trends at the two locations (Fig. 4C). However, pressure measured by the TriSonica mini is on average 3 hPa higher than the values measured by MURC. Constant bias corresponds to
about 13 meters higher elevation of MURC sensors compared to TriSonica. Wind speed and direction observations with the TriSonica mini and MURC (Fig. 5) showed qualitatively good agreement in both quantities throughout the campaign days. Differences in these quantities are subject to local wind conditions due to surrounding roughness elements. A true comparison of wind sensor performances must be made side-by-side as was done during the LAPSE-RATE campaign for wind sensors deployed on UAVs (Barbieri et al., 2019). Unfortunately, our surface module was not part of the inter-comparison.

**3.1.2   Aerosol particle concentration**

The combined FMI-KSU team was the only one to perform aerosol measurements during LAPSE-RATE. The aerosol number concentrations measured with the CPC of the surface sensor module were relatively stable during the whole campaign (July 15[th] -19[th]), and a similar diurnal cycle was observed every day (Fig. 6). The average total aerosol number concentration measured was around 1551 cm[-3], with minimum and maximum concentrations around 1211 and 2249 cm[-3], respectively, over the whole
campaign. A slight increase in aerosol concentration occurred during the midday hours, between approximately 16:00 UTC and 19:00 UTC (from 10:00 to 13:00 local time; Fig. 6). Increased anthropogenic activity in the San Luis Valley during morning hours is believed to result in an increase in production of primary particles from traffic and also creation of a pool of precursors for gas to particle conversion which activate due to daytime solar radiation. On average the total aerosol number concentration measured by POPS at the surface was about one third of that measured by CPC, around 532 cm[-3], with minimum and maximum
concentrations around 338 and 787 cm[-3], respectively, over three days of POPS measurements (Fig. 6). Occasional spikes in aerosol number concentration, up to 40 000 cm[-3] for CPC and up to 16 000 cm[-3] for POPS and lasting up to 3 minutes, were removed from datasets. Such high concentration peaks were caused by farm vehicles passing the sampling location.





The number concentration of particles measured by OPC-N2 (particles diameter between 0.38 to 17 μm) was generally very low, with an average concentration of about 1.2 cm$^{-3}$, a minimum concentration of 0.7 cm$^{-3}$ and a maximum concentration of
5.3 cm$^{-3}$ (Fig. 7). Coarse mode particle number concentration made up only a marginal fraction of the total aerosol number concentration. Corresponding average PM10 values were observed to be around 4.4 μg.m$^{-3}$, with a minimum 0.8 μg.m$^{-3}$ and a maximum of 127 μg.m$^{-3}$. Such total particle number concentrations and particulate matter mass concentrations are typical for rural areas, which the San Luis Valley undoubtedly is. Continuous air quality measurements are sparse throughout Colorado, and according to The Environmental Protection Agency (EPA) website an average PM2.5 concentration of 18±5 μg.m$^{-3}$, PM10
concentration 58±23 μg.m$^{-3}$ published as averaged maximum 24-hour concentration.

The particle number size distributions measured by POPS and OPC-N2 overlap well over eight size bins between 0.46 to 3.5 μm, and together both particle counters cover accumulation and coarse modes (Fig. 8). The diurnal changes in number size distribution were minimal and no condensation neither evaporation processes were observed.

### 3.2  Vertical profile observations

### 3.2.1  Reliability of observations

The measured profiles of temperature and relative humidity were divided between ascending and descending datasets by recorded altitude parameters. The data profiles showed quite a strong hysteresis in observed temperature and relative humidity, resulting in significant offsets between the ascending and descending profiles. Therefore, the observations are presented separately as solid curves for ascending legs and dotted curves for descending portions of the profiles (Fig. 9). The hysteresis is a
result of the flight strategy, based on which the goal was to reach as high altitudes as possible in a very short time. The BME280 Arduino sensor response was not fast enough to equilibrate to the quick changes in ambient conditions. Our ascent rates were approximately 5-8 and 3-5 m.s$^{-1}$ and descent rates were about 2-5 and 2-3 m.s$^{-1}$ for flights with particle module (FMI-PRKL1) and gas module (FMI-PRKL2), respectively. Based on our experience, the strong hysteresis disappears when the ascent and descent rates are close to 1 m.s$^{-1}$ or slower. The correction of hysteresis error was not made, since it is the coupled effect of the
wet and dry RH values and the elapsed time from the RH condition change during the fast vertical movement of the rotorcraft. The ascent data are probably closer to "true" profile and are preferred over the descend data.

Measured profiles of aerosol particles did not show any lag in instrument response during ascending and descending legs, and therefore both profiles were bin-averaged using 1 and 10 m vertical bins. Conversely, measured profiles of $CO_2$ revealed a slight lag in sensor response time, though both ascending and descending profiles were within the boundaries of measurement scatter.
Therefore, despite the apparent time lag, we proceeded similarly as in the case of aerosols, both ascending and descending profiles were averaged using 1 m and 10 m vertical bins.

Temperature and relative humidity vertical profiles showed expected behavior; a decrease in air temperature with altitude during the first three days, July 16$^{th}$-18$^{th}$. A temperature inversion was observed during the first two flights completed in the early morning of July 18$^{th}$. The temperature on the ground gradually increased due to increasing solar energy, resulting in the
gradual development of a convective boundary layer and a general warming of the environment near the surface. At the same





time, relative humidity was observed to generally increase with decreasing temperature as would be expected in well-mixed conditions, though occasional single layers were detected with the opposite behavior (Fig. 9). During the last flight day of the campaign, when the FMI-KSU team joined the common LAPSE-RATE mission to monitor cold air drainage, a temperature inversion was observed. During the first flight day (on July 16[th]) the relative humidity was not recorded since the calibration

constants were not read correctly.

### 3.2.2 New particle formation in the atmospheric vertical profile

Vertical profiles of particle number concentrations suggest that new particle formation (NPF) events were observed as an increase in particle number concentration in the nucleation mode (<20 nm) of the number size distribution. With the employed instrumentation suite, NPF can be monitored as a difference in particle number concentrations between readings from CPC1

(cut-off $D_{50}$= 7 nm) and CPC2 (cut-off $D_{50}$ = 14 nm). It should be noted that in some cases the CPC2 slightly over-counted particles compared to CPC1, resulting in negative $\Delta$CPC values. However, the difference in observed counts remains within the uncertainty of the instrument (<10%) at all times.

For the entire dataset, we have attempted to separate collected measurements into three different NPF classification regimes. These can be classified as: 1) No NPF event (July 17[th], see panel A in Fig. 10); 2) A weak NPF event (July 16[th], panel B in Fig.

10); and 3) A strong NPF event (July 18[th], Fig. 10 panel C). During July 16[th], NPF event developed throughout the day, with no NPF in the morning, a slight increase in particle number concentrations, a peak in NPF during midday followed by slight decrease in particle concentration towards the afternoon. NPF was observed at the surface throughout the observation period.

During July 18[th], NPF event was observed from early morning (8:03 local time) and continued until midday (10:59 local time), see panel C in Fig. 10. During this event, NPF took place only at high altitudes (about 3000 m MSL) and it was not

observable at the surface at all. In panels A-C of Fig. 10 the data only from ascending flights are shown for clarity since the descending rotorcraft would push aerosol particles downwards. This bias caused NPF events to be visible at much lower altitudes (about 2600 m MSL) compared to observations made during ascending flight. Previously, similar elevated layers of nucleation-mode particles have been observed in a number of different environments including for instance clean marine (e.g. Wehner et al., 2015), boreal forest (e.g. Leino et al., 2019) and other continental locations with varying amount of anthropogenic

influence (e.g. Wehner et al., 2010; Platis et al., 2016; Altstädter et al., 2018; Junkermann et al., 2018; Qi et al., 2019). Junkermann et al. (2018) linked their observations with elevated emissions from fossil fuel combustion. However, other sources likely contribute to particle formation in the cleaner environments.

To investigate the origin of air masses we ran the Flexpart dispersion model. The 96 h backwards footprint for July 18[th] at 20:00 UTC arriving to SLV at 1000 m AGL is presented in Fig. 11. On this day, the bulk of the air mass is transported from

275 West and was over the Pacific Ocean 72-96 h before arrival at San Luis Valley. A minor fraction of the air mass has continental origins between the measurement location and west coast of North America. A branch of this continental contribution covers the location of San Juan coal fired power plant approx. 200 km SW of the San Luis Valley (Fig. 11). Therefore, it is possible that the elevated layer of nucleation mode particles would originate in the power plant emissions. Decoupling of the layer





above 1 km from the surface can be observed as substantial drop in water vapor concentration at about 500 m AGL (see Fig.
10 panel C).

The presented observations on NPF support the following conclusions: 1) Precursor vapors can be distributed within mixing height of the atmospheric boundary layer resulting in NPF in deep vertical extend (July 16[th], Fig. 10 panel B). 2) In addition to vapor precursors, sink due to background aerosol for particles from NPF must be small enough. Levels of background aerosol concentrations are discussed below and support this conclusion. 3) Influence of long-range transport can be observed
in the vertical profile. Free tropospheric NPF is known to be a significant source of secondary aerosol over both marine and continental regions (e.g. Hermann et al., 2003; Benson et al., 2008; Merikanto et al., 2009). In such a situation, the surface area concentrations are usually observed to be very low (e.g. Twohy et al., 2002).

### 3.2.3   Accumulation and coarse mode aerosol particles

Vertical profiles measured with POPS up to 140 m AGL are presented as diurnal 1 m averages in Fig. 12. The profiles did
not show any variation with increasing altitude, the averaged particle concentration of 672 cm$^{-3}$ with minimum and maximum concentrations 500 and 1727 cm$^{-3}$, respectively. Flat profile concentrations correspond to those measured with POPS at surface. Similarly, averaged diurnal number size distribution did not go through any observable dynamical changes resulting from growth or evaporation of particles. The scatter in POPS data was caused by fast motion of rotorcraft when the ambient air was pushed against POPS' inlet and created pressure drop that internal pump did not handle very well, resulting in bias in POPS
measured volumetric flow.

Vertical profiles of particles measured by OPC-N2 did not show any significant variation with increasing altitude, and thus indicating a well-mixed boundary layer throughout the measurement column (Fig. 13). The averaged particle concentration was 5 cm$^{-3}$, similarly averaged PM1, PM2.5 and PM10 were 3.1, 4 and 9.7 μg m$^{-3}$, respectively. The main contribution to mass was made by aerosol particle load close to the surface. The result implies a lack of elevated plumes, despite the fact that there
were extensive forest fires in south Colorado before and during the LAPSE-RATE campaign.

### 3.2.4   Gas concentrations in the atmospheric vertical profile

Concentrations of $CO_2$ in the vertical profiles depend on both local sources and sinks as well as on atmospheric conditions including altitude, atmospheric mixing state and transport. The $CO_2$ and $H_2O$ vapor profiles showed higher concentrations only near the surface due to sources from soil and vegetation canopy respiration and evaporation. Averaged semi-diurnal (morning
to early afternoon) vertical profiles of $CO_2$ and water vapor during July 16[th] – 19[th] for both probes (Vaisala GMP343 and Licor LI-840A) are presented in Fig. 14. Even though both probes were calibrated in the laboratory at sea level with satisfactory results, they showed different absolute $CO_2$ concentrations in the field (SLV). The difference in $CO_2$ concentration increased with increasing altitude, and the GMP343 was found to measure $CO_2$ gradients independent of the time of day. This might be the result of imprecise pressure compensation for the GMP343 as provided by Vaisala (see the Section 2 for details). Since
we compensated GMP343 with temperature, RH and pressure obtained from BME280 sensor that showed of about +2 °C in temperature, -12% in RH and +2 hPa in pressure during the LAPSE-RATE inter-comparison measurements against MURC





(Barbieri et al., 2019), we also accounted for that bias in compensation resulting in an increase of $CO_2$ concentration by only 2 ppm. Very little variation in $CO_2$ concentrations was observed as a function of altitude on July 16[th] and July 17[th] (Fig. 14), indicating a well-mixed boundary layer. The same holds for water vapor concentration. However, on July 17[th] elevated surface concentrations were observed. During July 18[th] and 19[th], profiling started earlier in the morning (about 13 p.m. and 12 a.m. UTC, 7 and 6 a.m. local time, respectively) and we were able to observe gradients in $CO_2$ concentration between the rotorcraft and the ground level. These gradients extend to approximately 100 m from surface on July 18[th] and 250 m on July 19[th]. Similarly for water vapor concentration, on July 18[th] and 19[th] the gradient extends the whole profile.

Figure 15 demonstrates a notable change in $CO_2$ concentration with increasing altitude on July 19[th], where the first five flights from 11:44 to 13:41 UTC show a near-surface $CO_2$ gradient. This gradient is detected to include a decrease of approximately 30 ppm (450 to 420 ppm for the LI-840A, and 430 to 400 ppm for the GMP343) in first 350 m above surface. During the subsequent five flights (14:11 to 16:42 UTC, 8 and 10 a.m. local time), the $CO_2$ concentration profiles began to be more uniform all the way to the surface, with hints of a slight negative gradient observed with the LI-840A. A negative gradient was not observed with GMP343 sensor, probably due to its lower sensitivity. Such a negative gradient could be explained by ground vegetation photosynthetic uptake and the vertical transport of $CO_2$ to higher altitudes, as reported e.g. by Li et al. (2014). However, no ground flux measurements were present during the campaign to support this hypothesis. Water vapor concentration showed an opposite trend, with the profile being flat in the early morning (about 7 g.kg$^{-1}$) and the concentration at the surface gradually increasing up to 8.5 g.kg$^{-1}$, and at 350 m above the surface it was 6.75 g.kg$^{-1}$ for the last flight at 16:42 UTC.

Unfortunately, we were not able to acquire vertical profiles or ground level concentrations of $NO_2$, $O_3$, CO and $SO_2$ with the Vaisala AQT400 sensor. All data collected during LAPSE-RATE were far below the manufacturer declared detection limits. The AQT400 sensor is rather suitable for detection or identification of sources with higher concentrations, though this is not supportive of detailed measurements in clean environments with very low background concentrations of both gases and particulate matter. Furthermore, the AQT400 standard calibration is valid for the range 800 - 1200 hPa, and given the elevation of the San Luis Valley (about 2300 m MSL) the ambient pressure during the campaign was around 780 hPa at the surface, impacting the accuracy of the calibration significantly.

## 4    Concluding remarks

Flight teams from the Finnish Meteorological Institute and the Kansas State University participated in LAPSE-RATE campaign in Sun Luis Valley, CO, USA during July 14[th] - 19[th], 2018. The teams were operating from the same location and created a sound dataset on the properties of aerosol particles and gases in the atmospheric vertical profile. During the campaign the FMI modular system with off-the-shelf instrumentation was tested.

From deployment of UAS-borne off-the-self sensors in a high-altitude environment we learned that particle counters used during the LAPSE-RATE campaign worked flawlessly together. Even though they were deployed on different platforms the obtained number size distributions overlapped well over several size bins. The particle counters together covered aerosol



particles ranging between 7 nm to 18 µm in diameter, however with a gap in the size range between 15-130 nm. Such a setup in principle allows very affordable observations of new particle formation events (nucleation mode), particle growth by condensation and coagulation (accumulation mode), and dust episodes (coarse mode) in vertical profile. We also learned that the Vaisala GMP343 $CO_2$ concentrations and vertical profiles do not match Licor Li-840A after post-processing the data due to not very precise pressure correction provided by manufacturer. The precision and response times of Li-840A remain better.

Water vapor content profiles obtained by Li-840A compared well against those calculated from radiosonde profiles. The air quality sensor Vaisala AQT400 is more suitable for environments with elevated levels of gas pollutants, in clean background location like SLV the measured values remained lower than the manufacturer stated detection limits.

During the short measurement campaign features of aerosol particles, $CO_2$ and water vapor specific to the Sun Luis Valley were acquired. As an example, a new particle formation event at altitudes at and above 700 m above the surface, decoupled

from the ground, was observed. This is an observation that could not easily be obtained by known remote sensing methods, but is easily collected using airborne in-situ methods.

*Data availability.*  All of the LAPSE-RATE campaign datasets including FMI-KSU have been uploaded to a LAPSE-RATE community set up at the Zenodo data archive https://zenodo.org/communities/lapse-rate/

*Author contributions.*  D.B., A.H., G.B., O.K. and V.V. writing, reviewing and editing of manuscript, D.B., J.G., O.K. conducted the mea-

surements. D.B. made dataset analysis and validation. V.V. made Flexpart analysis.

*Competing interests.*  The authors declare no conflict of interest.

*Acknowledgements.*  Authors would like to acknowledge following foundation for providing financial support of the project: KONE foundation, ACTRIS-2 - the European Union's Horizon 2020 research and innovation programme under grant agreement (No 654109), ACTRIS PPP - the European Commission under the Horizon 2020 – Research and Innovation Framework Programme, H2020-INFRADEV-2016-

2017 (Grant Agreement number: 739530), Academy of Finland Center of Excellence programme (grant no. 307331) and the US National Science Foundation CAREER (1665456). In addition, Limited general support for LAPSE-RATE was provided by the US National Science Foundation (AGS 1807199) and the US Department of Energy (DE-SC0018985) in the form of travel support for early career participants. Support for the planning and execution of the campaign was provided by the NOAA Physical Sciences Division and NOAA UAS Program Office. Finally, the support of UAS Colorado and local government agencies (Alamosa County, Saguache County) was critical in securing site

permissions and other local logistics. D.B. and J.G. would like to especially acknowledge Dave L. Coach for acting as a Pilot-In-Command for FMI team. Handix Scientific, LLC is acknowledged for providing their POPS instruments for the campaign at no cost.





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

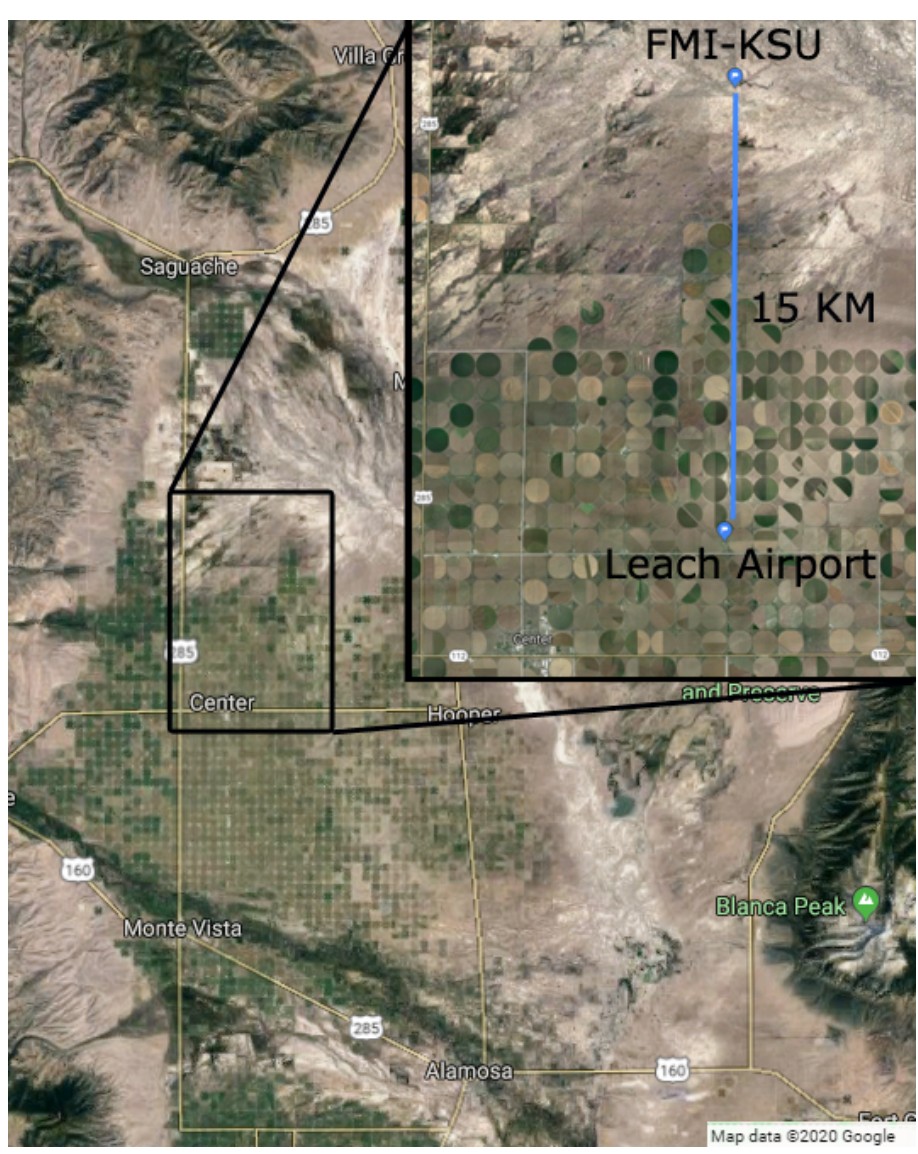

**Figure 1.** Map of wider San Luis Valley, the insert shows location of FMI - KSU team spot, and distance from Leach Airport. The airfield is located approximately 3.2 km ENE of the commercial district of Center, Colorado and 32 km NNW of Alamosa, Colorado. Background map courtesy of ©Google Maps.



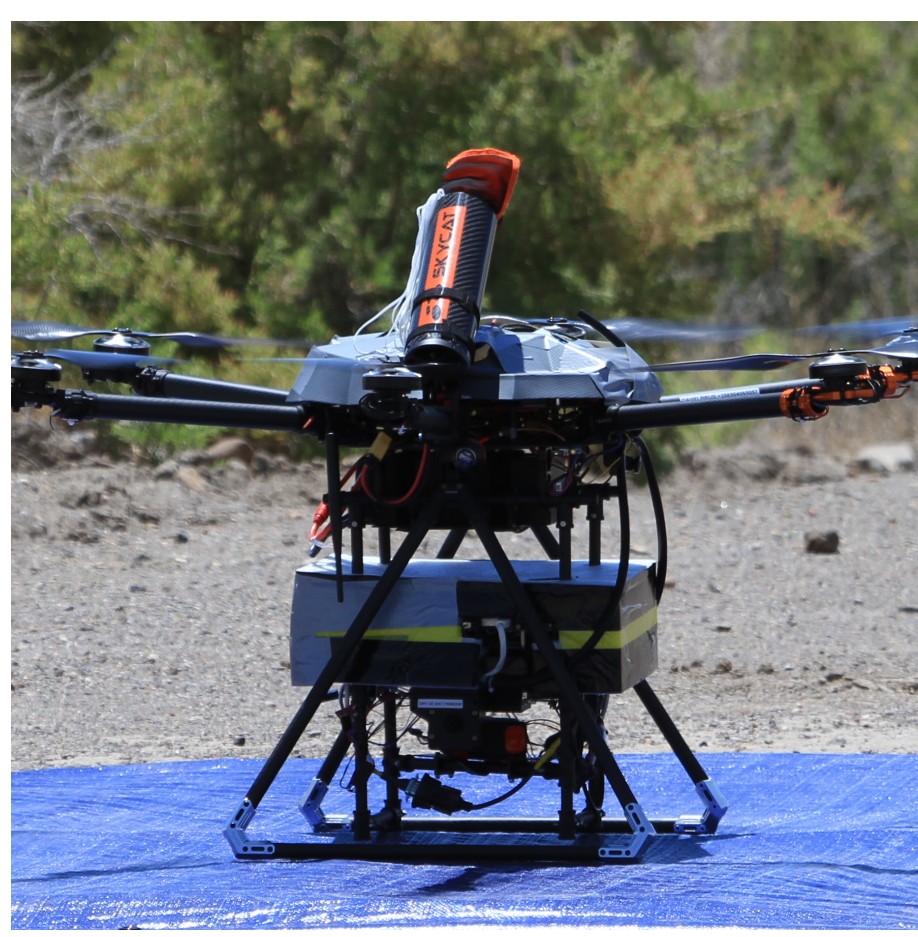

**Figure 2.** Particle module attached to Tarot X6 hexacopter at the take-off/landing spot - blue tarp.





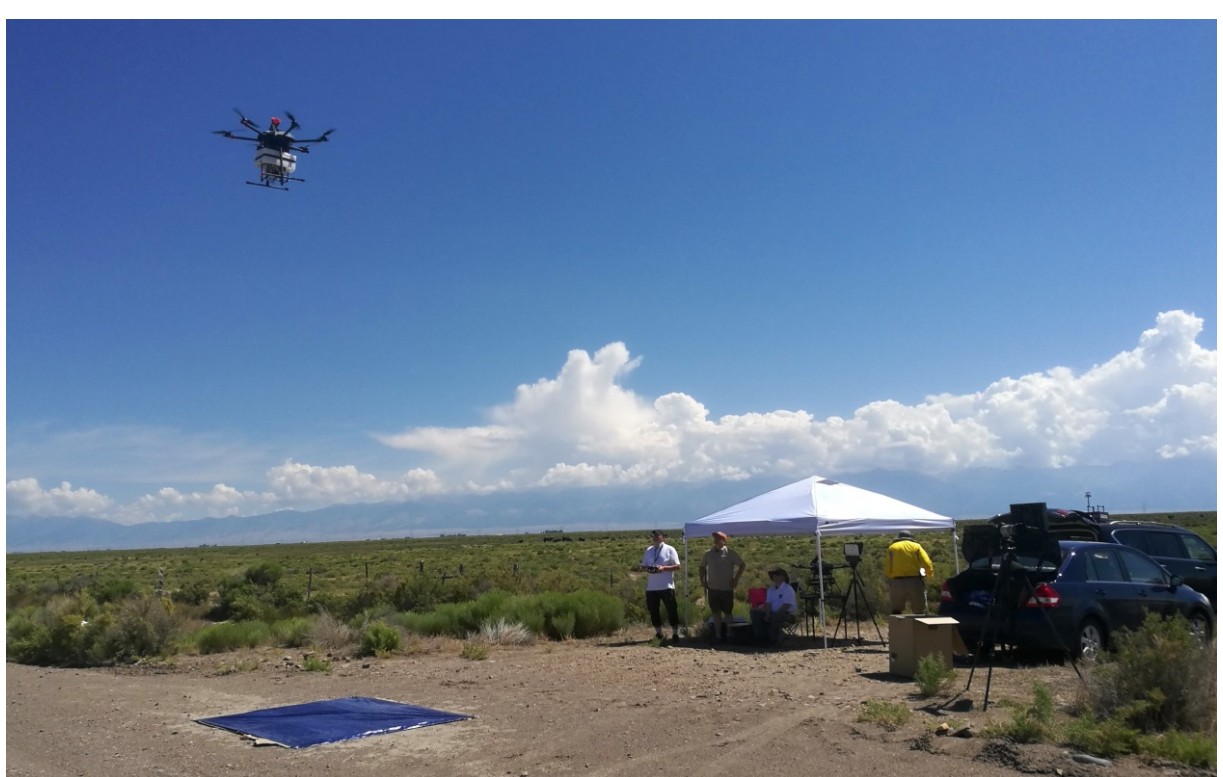

**Figure 3.** FMI-KSU team at dedicated spot during LAPSE-RATE campaign, gas module attached to Tarot X6 hexacopter during airborne operation. Ground module with TriSonica mini can be seen on the roof of SUV.

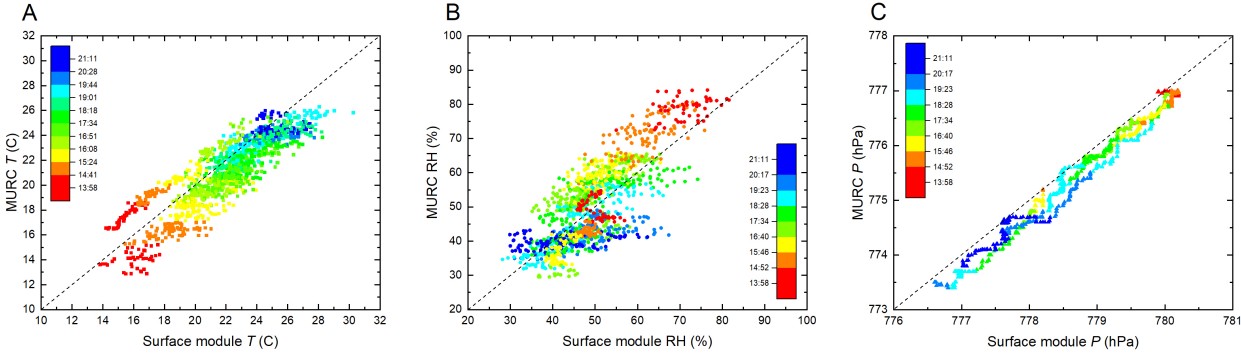

**Figure 4.** Diurnal values measured with TriSonica mini mounted to FMI surface module compared to observations by MURC tower positioned at Leach Airport A) temperature, B) RH and C) pressure.

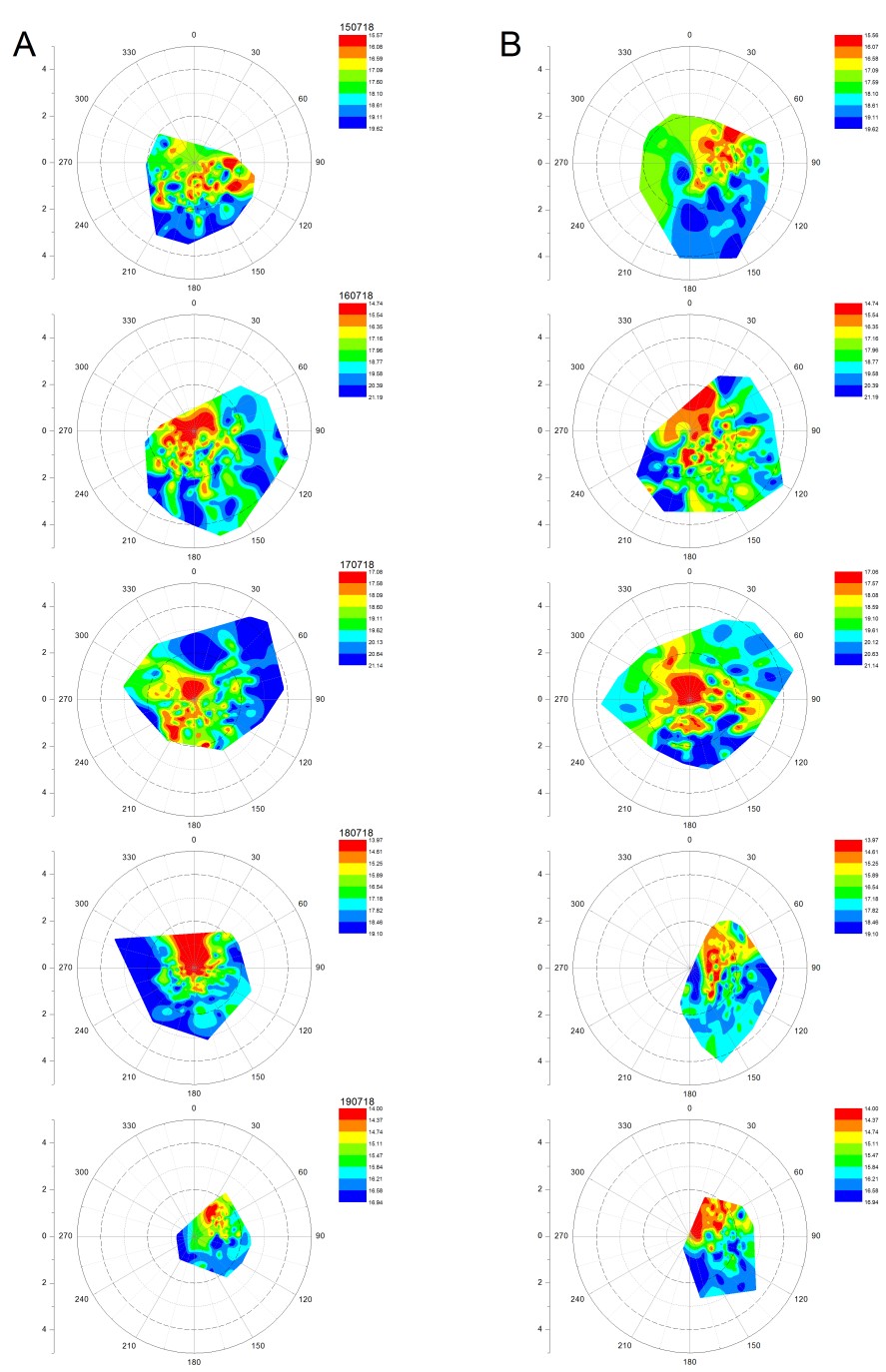

**Figure 5.** Diurnal contour wind roses as a function of time of day (UTC time), A) TriSonica mini WS and B) MURC.



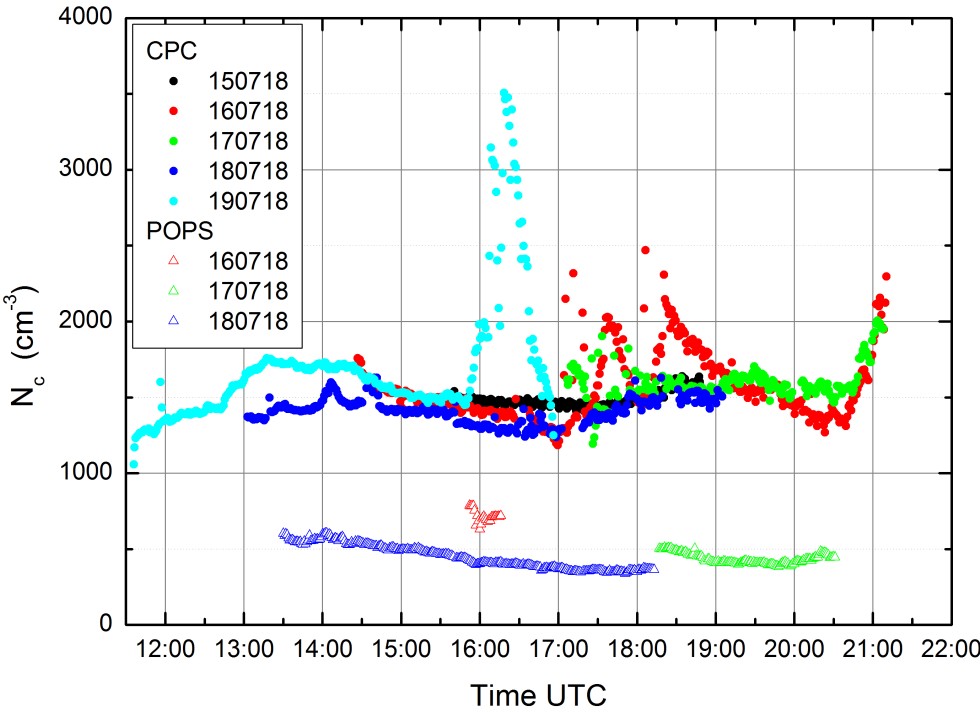

**Figure 6.** Total aerosol number concentration (minute averages) measured with CPC TSI 3007 (from July 15th to 19th) and POPS (from July 16th to 18th) at the surface.

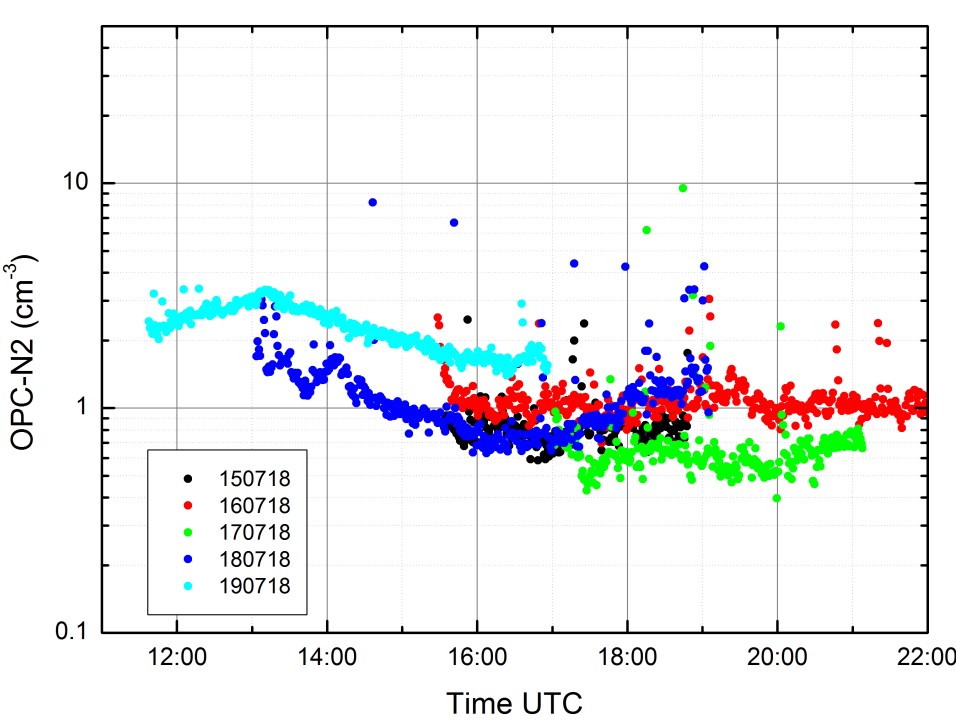

**Figure 7.** Total aerosol number concentration (minute averages) measured with OPC-N2 at the surface from July 15[th] to 19[th].



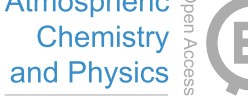

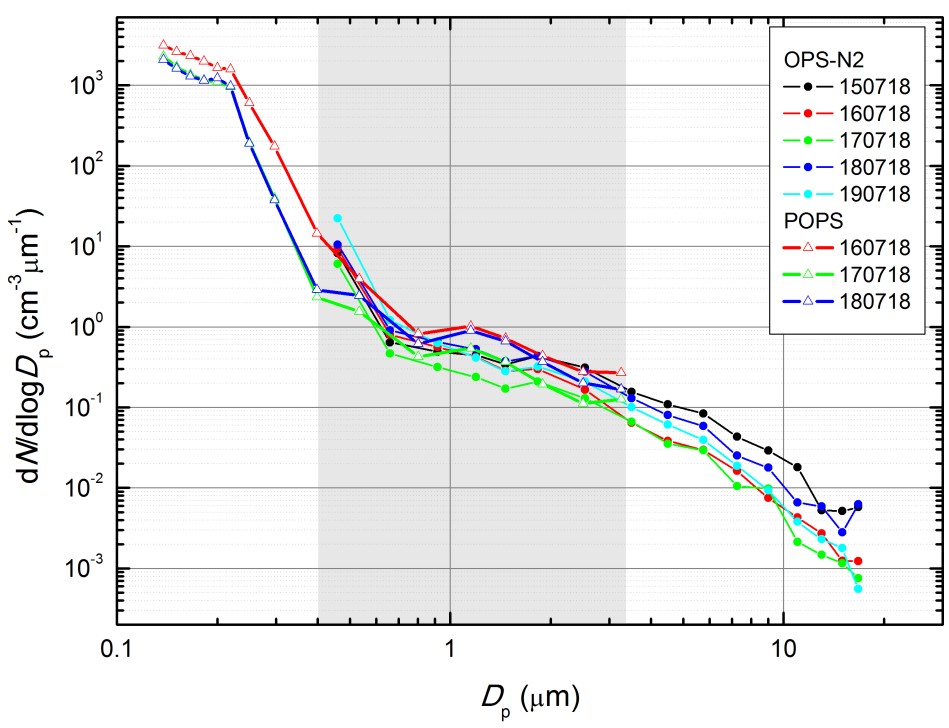

**Figure 8.** Aerosol number size distribution (diurnal averages) measured with OPC-N2 and POPS (plotted separately) at surface from July 15[th] to 19[th]. The grey area represents overlapping sizes of bout both particle counters.





**Figure 9.** Vertical profiles of temperature and RH measured by BME280 Arduino sensor from July 16th – July 19th, solid lines–ascend data dotted lines–descend data

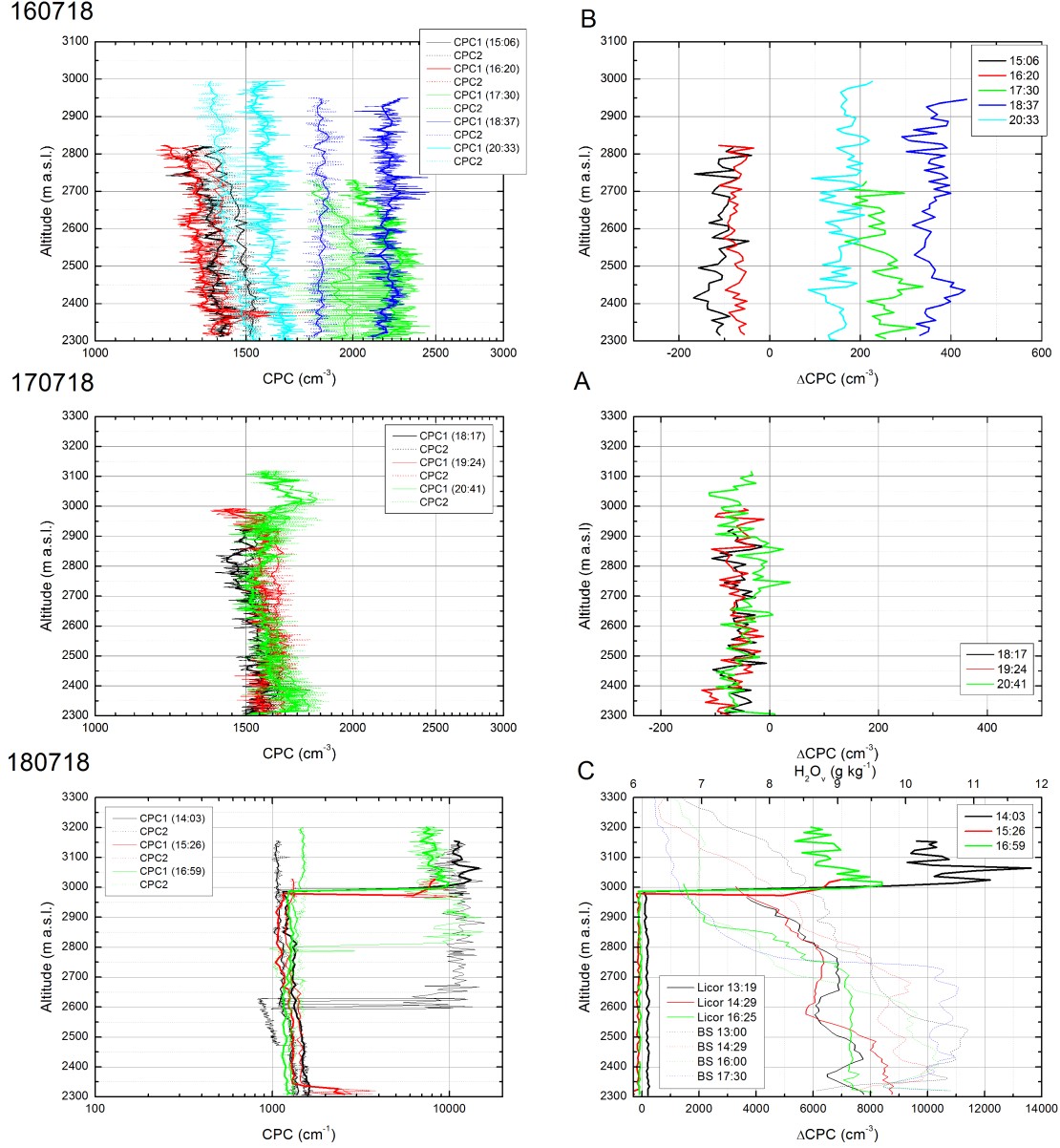

**Figure 10.** Vertical profiles of total aerosol measured with two CPC TSI 3007 in particle module during July 16[th]-18[th], CPC1 with cut-off diameter $D_{50}$ = 7 nm (solid lines) and CPC2 with $D_{50}$=14 nm (dotted lines). Observation of new particle formation (NPF) event: A) no NPF event measured on July 17[th]. B) A weak NPF event measured throughout the measurement heights on July 16[th], the event developed throughout the day with no NPF in the morning, followed by a slight increase in particle freshly formed particle number concentration during midday and decreasing during the afternoon. C) A strong NPF event was observed on July 18[th] from 3000 m MSL upwards through early morning (8:03 local time) and midday (10:59 local time). Water concentration vertical profiles measured with Licor LI-840A and calculated from balloon sounding's parameters demonstrates the decoupling of free troposphere nucleation from the atmospheric boundary layer.



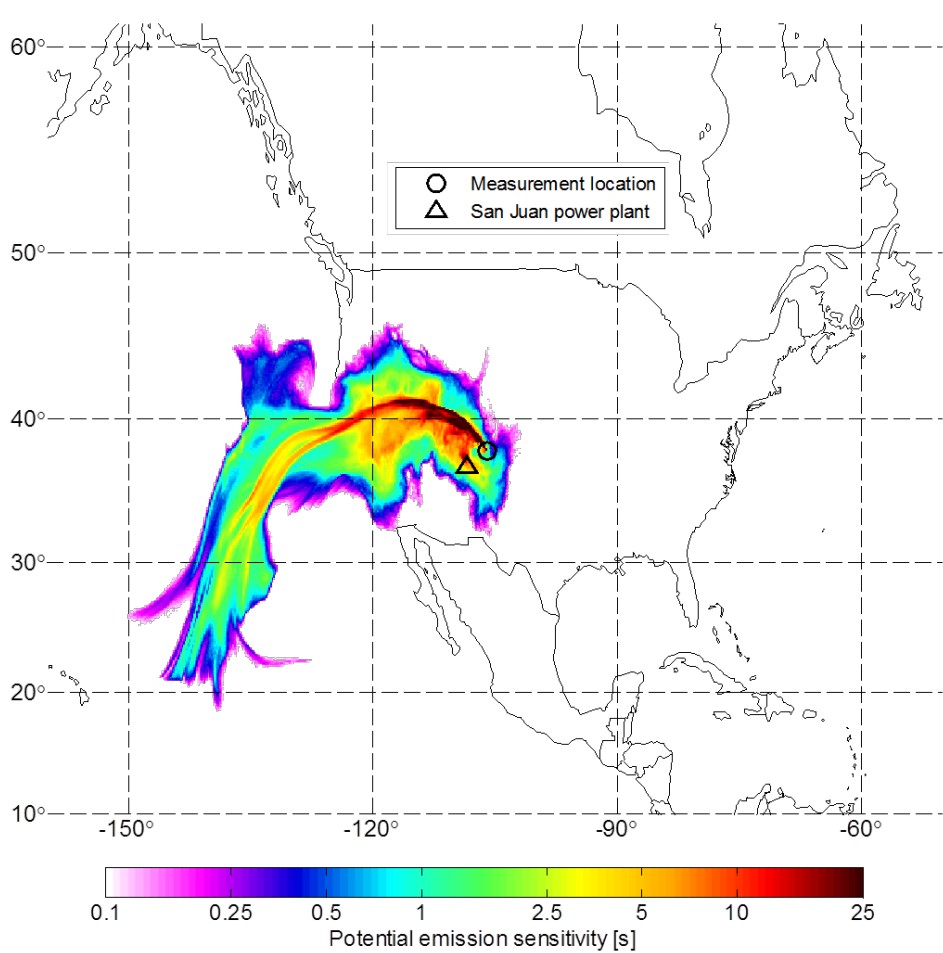

**Figure 11.** Emission footprint calculated by Flexpart dispersion model for July 18$^{th}$ 96h backwards, arriving at 20:00 UTC at 1000 m AGL.





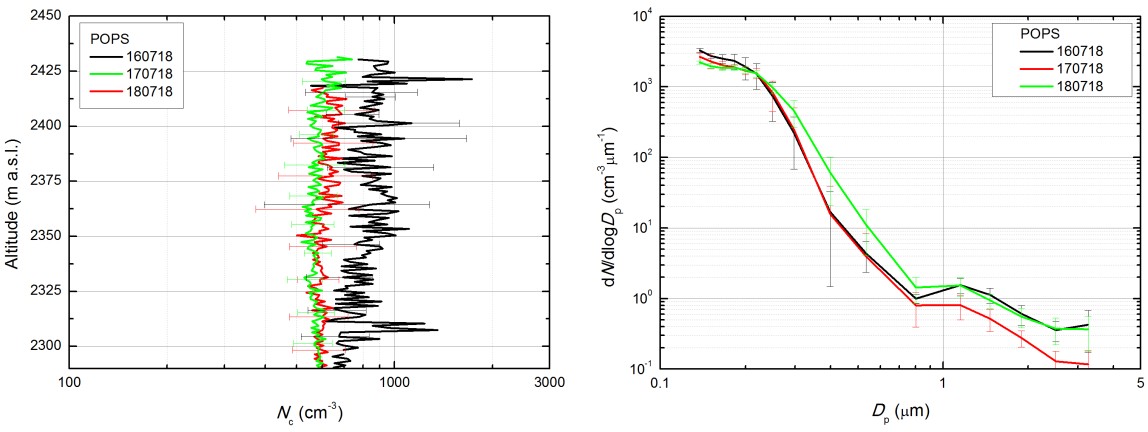

**Figure 12.** Daily averaged vertical profiles of total particle count and number size distributions measured with POPS, from July 16[th]-18[th].

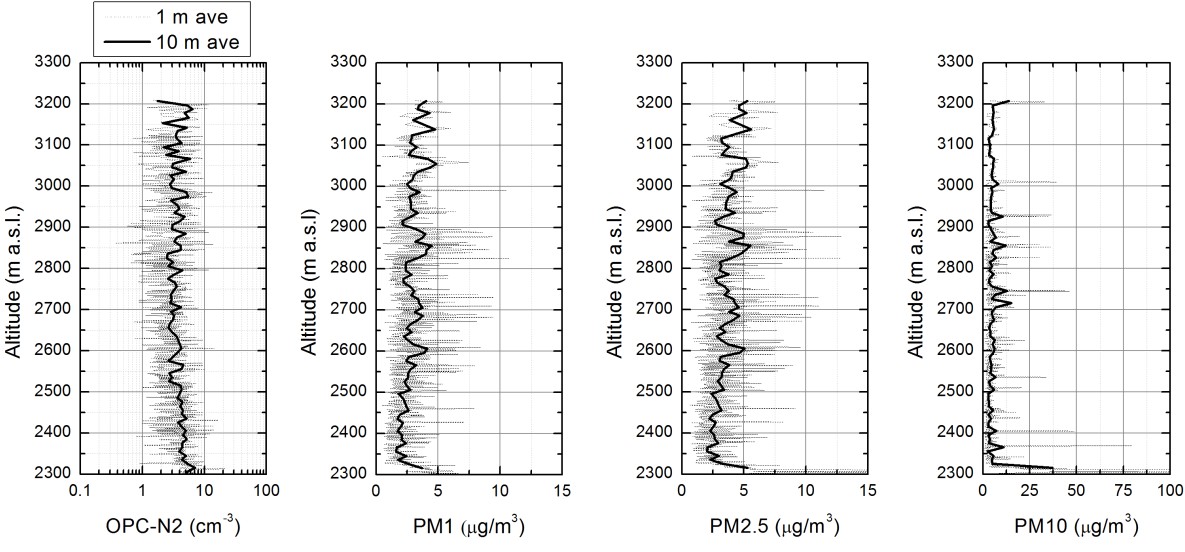

**Figure 13.** Averaged vertical profile OPC-N2 coarse particle total count - black line from July 16[th]-18[th]. Averaged vertical profiles of PM1, PM2.5 and PM10 values measured with OPC-N2-black line, from July 16[th]-18[th].

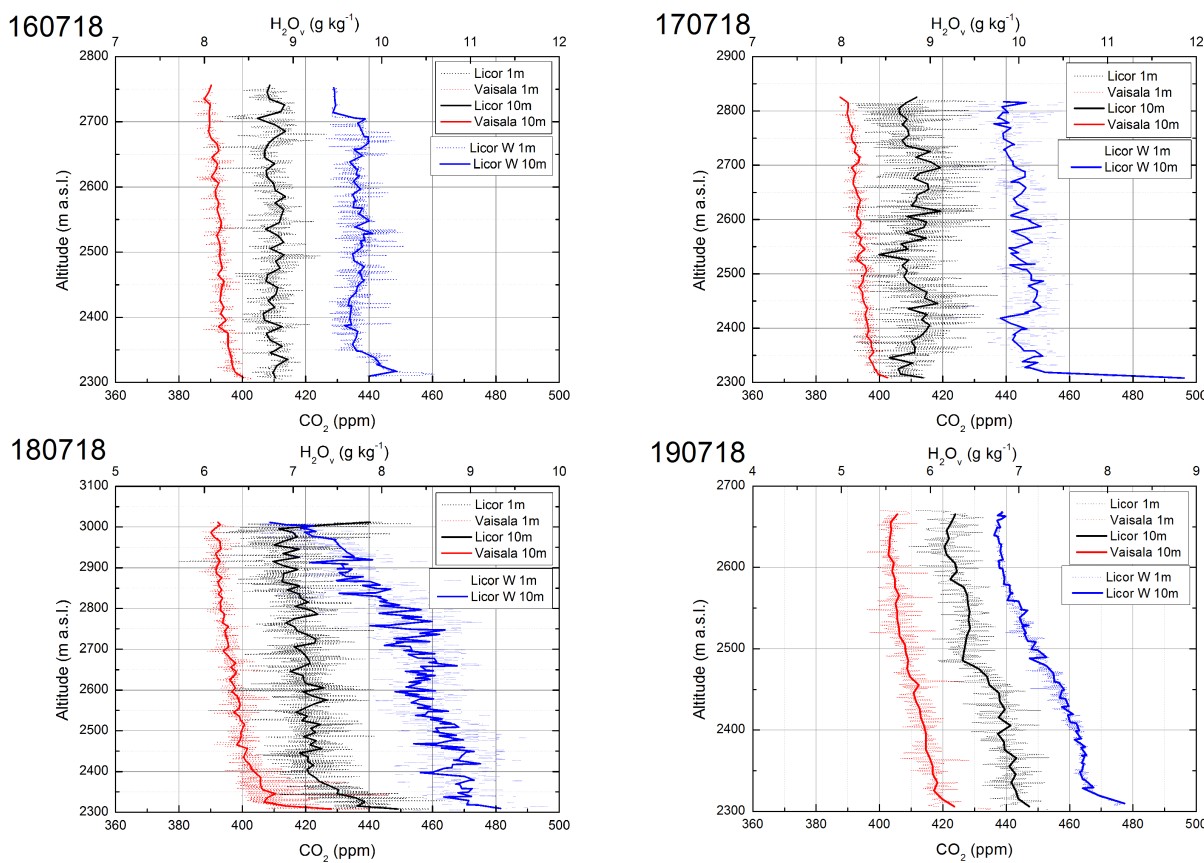

**Figure 14.** $CO_2$ and water vapor vertical profiles measured with Licor LI-840A and Vaisala GMP343 on July 16th-19th. Data plotted as 1 meter (dotted lines) and 10 meter averages (solid lines). Please note different altitude on daily figures.



**Figure 15.** $CO_2$ and water vapor vertical profiles measured with Licor LI-840A and Vaisala GMP343 on July 19th. Early morning data measured with LI-840A show visible gradient with higher $CO_2$ concentrations at the ground level, the profile flattens with increased Sun activity. The profiles measured with Vaisala GMP343 show weaker gradient because of its lower sensitivity. Water vapor profiles were firstly flat with gradient developed with time due to increased solar radiation.