# Peer review of "Measurement report: Properties of aerosol and gases in the vertical profile during the LAPSE-RATE campaign"

_Atmospheric Chemistry and Physics, 2020_

## Referee Comment (RC1) · Anonymous Referee #1 · 10 Jul 2020

Review of a manuscript titled with "Measurement report: Properties of aerosol and gases in the vertical profile during the LAPSE-RATE campaign" by David Brus, et al. 2020

General comments: This paper demonstrates the unique aerosol and gas measurements via UASs and provides a promising data example for environmental research. The authors presented a very useful payload for the atmospheric study and shared the exciting datasets from July 14 – July 18, 2018. However, analysis of the measurements is limited, and lack of a meaningful uncertainty estimation, which is critical for many data applications, such as the modeling evaluation.

[Figure]

Specific comments: P3, section 2.1: What is the typical flight operation? "The maximum endurance of these rotorcraft was about 15 minutes." (in line 61) Does that suggest the measurements with this platform only last for 15 mins? If so, it is too short for any study. Maybe the authors can explain how to operate this platform to provide meaningful vertical datasets?

P4, section 2.2, line 89, is this "basic meteorological sensor" the Arduino Bosh BME280 sensor?

P4, section 2.2, There is no uncertainty or accuracy information about the CPCs and OPC. The BME280 sensor accuracy mentioned in this section (in line 100) are all based on manufacturer statements, and very different from the comparison difference discussed in line 106-107. Does the field environment affect those manufacturer's accuracy? Were other sensors compared with high precision "siblings" in the laboratory, like CO2 sensors? If so, that information should be included here too. Does the KSU Matrice 600 Pro contain any other met sensors? POPS was not shielded from direct sunlight during the flights. Does that cause any overheating? What is the optical chamber temperature during the flights?

P6, section 3.1.1, the ground meteorological comparison is meaningless because two sites are 15 km apart. Maybe change the Fig 4 (Surface vs. MURC) to diurnal variation plot (time vs. T, RH, P).

P7, section 3.1.2, if I understand the section 2.2 correctly, a duplicate CPC, OPC, and POPS were operated on the ground. Do you compare them with the flying version on the ground?

Figure 8, "POPS and OPC-N2 overlap well over eight size bins" may overstate the comparison. The figure is in log scale, and POPS seemed to be a factor of 2 of the OPC-N2.

P8, section 3.2.1, The BME280 sensor has a +2 C difference comparing to the MURC.

Will this difference constant for the hysteresis temperature profile? Will the observed T and RH data scientifically useful?

P9, section 3.2.2, it is great to see the new particle formation detection capability developed here. It will be beneficial to correct the over-counting behavior of CPC2 before calculating the delta(CPC). Line 266, if the descending rotorcraft would push aerosol particles downwards. Would the ascending flight push aerosol particles upwards? What is the ascending and descending rate of this flight?

Line 273, the site is about 700 m AGL. Will the 1000 AGL Flexpart dispersion model represent the 700 m AGL condition?

P10, section 3.2.3, It sounds that the platform motion has an impact on the POPS data. Please quantify the impactor. Does the author characterize the inlet loss for all the aerosol instruments – CPC, POPS, and OPC? It is critical to know the inlet loss for OPC during the descending and ascending because the concentration is very low – 5 cmˆ-3.
* * *

---

## Referee Comment (RC2) · Anonymous Referee #2 · 11 Aug 2020

The manuscript "Measurement Report: Properties of aerosol and gases in the vertical profile during the LAPSE-RATE campaign" by Brus et al. provides a description of UAS platform measurements of particles, trace gases and meteorological parameters in the San Luis Valley of Colorado in July 2018. The measurements were made as part of a multi-institution campaign associated with the ISARRA meeting that was held in Colorado that summer. The authors provide substantial details regarding the instruments, platforms and operation, and as such, the manuscript serves to provide a solid example of the kind of research that can be accomplished, and likely most effectively, using sUAS platforms. The manuscript is well organized and fairly well written, but the readability would benefit from careful copyediting for usage and punctuation. Overall,

given the focus of the paper on the description of measurements and measurement systems rather than scientific analysis, I would think the manuscript would be a better fit in AMT than ACP.

General comments:

The authors should be consistent with the description of the team (FNMI-KSU) or teams (FNMI and KSU)—both are used in the manuscript, e.g. L42 "A flight team", L44 "by the FMI and KSU teams"

In terms of lower atmosphere/boundary layer studies, height AGL is more relevant or important to know than absolute altitude (MSL). Similarly, for diurnal processes such as boundary layer development, using local time is clearer than UTC, which requires the reader to calculate the local time to interpret the data.

The discussion of the vertical profile observations is mostly descriptive rather than analytical, with the exception of the final paragraph of 3.2.2 (NPF), which would still benefit from substantially stronger arguments to derive the conclusions from the observations.

Figures 4 and 5 are somewhat tangential to the focus of the manuscript. The inclusion of the descent data in Figure 9 clutters the picture when it has been argued in the text that the ascent data are believed to be more reliable. A single panel with both ascent and descent for comparison could be included instead to illustrate the description in the text.

Specific comments:

L14: The opening sentence of the introduction is a bit absolutist. Perhaps "Most air pollution, including both gases and particulates, is released near the surface from various sources." The specific mention of wet deposition seems out of place since it is discussed nowhere in the manuscript—it could be omitted.

L18: The sentence "The atmospheric boundary layer can be investigated. . ." lists many measurement methods and relevant references, but then just stops without any following statement about why this list was presented. UAS are merely included as one item instead of the paragraph being used to set up the paper by describing why one might use sUAS instead of, or to complement, other methods.

L46: suggest replacing "leveraging" with "utilizing" or "using"

L72: "employed a parachute system"—was the parachute actually used regularly for landing, or was it there in case of emergency (power failure) and not actually needed during the campaign?

L79: If 15.5 kg GTW / 5.5 kg PW would have been too much for operation at the altitude of the SLV, what were the actual KSU Matrice operational weights for LAPSE-RATE?

L85: "These modules were easily detached...swapping between sensor modules"—but since you were using two aircraft, did you actually swap the payloads? It didn't seem so from the description of operations.

L90: "The CPCs were calibrated"—perhaps "were set", "were adjusted"

L92: The voltage applied to the TEC (or Peltier cooler) sets [or regulates], the temperature difference between the saturation and condensation regions

L94: given the cut-off (50%?) diameters of 7 and 14 nm how did the temperature instability limit the detection of sub-20 nm particles—how much did the cut-points vary? Was this incorporated in uncertainty, or was the delta-T measured and compensated?

L96: "The platform was enclosed on all sides except the bottom with polylactic acid (PLA) foam to shield"; "Bosh" → "Bosch"; "pressure (P), temperature (T) and relative humidity (RH).

L100: "6 ms and $\pm 1$ hPa for P, 1 s and $\pm 0.5$ °C for T, and 1 s and $\pm 3\%$ for RH."

L120: "oxygen"? Assumed to be $\sim 20\%$ of P?

L128: suggest stating here that the AQT400 was not useful in the SLV environment.

L146: How long was the horizontal inlet?

L147: What was the nature/purpose of the POPS "custom electronics"?

L149: What is the expected effect of the POPS not being shielded from the sun over the duration of a profile? Did you evaluate the effect by running the instrument on the ground for the same period to see if a measurable effect occurred?

L163: Typically, MSL would be denoted "altitude" and AGL "height".

L178: "A detailed description of the daily meteorological conditions in the SLV during the LAPSE-RATE campaign"

L181: The data presented in figures 4-7 seem to indicate that the surface measurements were not continuous, but only occurred during portions of each day

L219: "according to the US Environmental Protection Agency"; EPA or USEPA are not needed since there is no further reference in the paper

L230: "based on which the goal"—perhaps "the goal of which was to reach the highest altitude possible in a very short time."

L256: From figure 10 it appears that the bias between the two CPCs was fairly systematic, raising a question about either the determination of cut-off diameter or relative sampling efficiency. This potentially affects quantitative analysis of the results, but does not necessarily impact the qualitative conclusions drawn

L273: The discussion of the Flexpart back trajectories and interpretation of the resulting footprint (shown in Figure 11) is rather superficial and speculative

L293: The description of the effect of aircraft motion on the POPS sampling is somewhat vague. How did the motion cause the effect? Was it the same for the horizontal and vertical inlet configurations (L145-147)?

L298: "was 5 cm-3, similarly averaged PM1"—should either use a ";" or ", while"

L304: "evaporation"—"transpiration" if just from plants or "evapotranspiration" if from the environment (surface + plants)

L310: "from a BME280 sensor that showed a bias of about..."

Figure 10: Suggest labeling the panels in order (A, B, C) and change the references in the text to match

---

## Author Comment (AC1) · 9 Oct 2020

We thank reviewer #1 for constructive comments, we really appreciate their time spent reading our manuscript.

*General comments: This paper demonstrates the unique aerosol and gas measurements via UASs and provides a promising data example for environmental research.*

*The authors presented a very useful payload for the atmospheric study and shared the exciting datasets from July 14 – July 18, 2018. However, analysis of the measurements is limited, and lack of a meaningful uncertainty estimation, which is critical for many data applications, such as the modeling evaluation.*

RC1) *Specific comments: P3, section 2.1: What is the typical flight operation? "The maximum endurance of these rotorcraft was about 15 minutes." (in line 61) Does that suggest the measurements with this platform only last for 15 mins? If so, it is too short for any study. Maybe the authors can explain how to operate this platform to provide meaningful vertical datasets?*

AR1) For the LAPSE-RATE campaign we purchased a brand-new set of LiPo batteries, since it was the only way how to secure our operation. There was not enough time to make enough of charge-discharge cycles to rich the full batteries capacity. We put a lot of emphasis on safety during our operation, when battery voltage reached 23 V (6S 1600 mAh) during the rotorcraft ascend we started the descend to safely land. The progress in achieved maximum altitude is clearly visible from FIG 9 in the manuscript. If well cycled battery is used, the flight times are about 20 minutes, depended on environment (wind and temperature), our maximum achieved height ever with this rotorcraft was 1000 m AGL when flying in stabilized mode and conditions were perfect. Usually the combination of legal flight limits and the rotorcraft capabilities dictate the flight strategy. For example, we can choose, when measuring aerosols, do we want to characterize whole vertical column within the legal limits (3000 feet) but with poor statistics or rather have better statistics and give up on full height of the column. We made our choice and were able to observe high altitude NPF event.

We will elaborate this issue in revised manuscript.

RC2) *P4, section 2.2, line 89, is this "basic meteorological sensor" the Arduino Bosh BME280 sensor?*

AR2) Yes, we meant Arduino Bosh BME280 to be basic meteorological sensor. It will be clarified in revised version of the manuscript.

RC3) *P4, section 2.2, There is no uncertainty or accuracy information about the CPCs and OPC. The BME280 sensor accuracy mentioned in this section (in line 100) are all based on manufacturer statements, and very different from the comparison difference discussed in line 106-107. Does the field environment affect those manufacturer's accuracy? Were other sensors compared with high precision "siblings" in the laboratory, like CO2 sensors? If so, that information should be included here too. Does the KSU Matrice 600 Pro contain any other met sensors? POPS was not shielded from direct sunlight during the flights. Does that cause any overheating? What is the optical chamber temperature during the flights?*

AR3) The uncertainty and accuracy information on CPCc, OPC and BME280 will be elaborated in revised manuscript.

In Barbieri et al. 2019 it was discussed that placement of the sensor (shielded/not-shielded, aspirated/not-aspirated) plays dominant role rather than its own accuracy and uncertainty, that was also valid in our case, sensors shielded but not forcefully aspirated. When BME280 sensors are calibrated in environmental chamber – well controlled conditions, against national standard at FMI, their response is very satisfactory, please see FIG 1 attached.

[Figure]

Figure 1. T and RH calibration for 9 pcs of BME280 sensors against FMI national standard.

The KSU Matrice 600 Pro did contain Meteorological sensor (iMetXQ2, International Met Systems, Grand Rapids, MI, USA) borrowed from Oklahoma State University. However, the sensor data were found corrupted, and we were not able to recover the files for this manuscript.

The flight POPS was in direct sunshine only for a limited time due to the flights being short and was being cooled by an airflow during the flights. As a consequence, the highest optical chamber temperature recorded was 33C. The ground POPS was exposed to longer durations of direct sunshine and less airflow. As a result, the highest optical chamber temperature recorded was 53C. The manufacturer does not give a specific recommendation for the optical chamber temperature, but after reviewing the data raised no concerns about the overheating. As such, we believe that the temperatures are within acceptable ranges, especially for the flight unit.

*RC4) P6, section 3.1.1, the ground meteorological comparison is meaningless because two sites are 15 km apart. Maybe change the Fig 4 (Surface vs. MURC) to diurnal variation plot (time vs. T, RH, P).*

AR4) In general, during LAPSE-RATE campaign there was a lack of reference measurements. We will change the Fig. 4. To diurnal plot as suggested by reviewer, but we would like to keep the MURC data too even though they were measured at different height and 15 km distance.

*RC5) P7, section 3.1.2, if I understand the section 2.2 correctly, a duplicate CPC, OPC, and POPS were operated on the ground. Do you compare them with the flying version on the ground?*

AR5) Yes, we performed a short comparison (about 5 minutes) before each flight among the particle counters to check their performance, mostly visually from the laptop screen, if the number concentrations roughly correspond to each other. The rotorcraft with particle module was not in the same location as surface module, neither the particle counters were using the same inlet. The rotorcraft was standing on the camping table approximately one third of the height of surface module which was placed on the car roof. Since the provided comparison is rather semi quantitative.

The comparison of rotorcraft particle module to surface module for CPC could be seen in attached FIG. 2, we must point out that each CPC was calibrated to different D50 cut-off, the most pronounced disagreement could be seen on Jul 16th when a weak NPF took place also at the surface (the red circles).

[Figure]

Figure. 2 Inter-comparison of CPCs mounted on rotorcraft particle module (CPC1 and CPC2) and surface module (CPC Ground).

The comparison of OPCs in particle and surface module is shown in attached FIG. 3. In some cases, the OPC on particle module shows higher concentrations than the OPC on surface module. This might due to rotorcraft proximity to dusty surface during comparison. Similarly, when we compare normalized concentration per bin, the OPC on particle module slightly overcounts in all bins, see FIG. 4.

[Figure]

Figure 3. Daily comparison of total number concentration of OPCs mounted on rotorcraft particle module (OPC-N2_RC) and surface module (OPC-N2_Surface).

[Figure]

Figure 4. Daily comparison of normalized concentration per bin of the OPCs mounted on rotorcraft particle module (OPC-N2_RC) and surface module (OPC-N2_Surface).

There was no intentional comparison made for the pair of POPS counters, however we made a comparison of total particle number concentration using the air unit data just before the flight, when the KSU rotorcraft was ready for take-off, e.g. height was zero or close to zero, see Fig 5. The particle concertation data are slightly biased toward higher counts of air unit, on average about 10%.

[Figure]

Figure 5. Comparison of POPS air and surface unit particle number concentration.

We will add following to manuscript: "A short comparison (about 5 minutes) was performed before each flight among the particle counters to check their performance, mostly visually from the laptop screen Based on data postprocessing, the CPCs of particle and surface modules compared well within the manuscript stated uncertainty of 10%, except for July 16[th] when NPF at the surface level took place. This is due to different calibrated cut-off diameter of each CPC. The OPCs compared within factor of 2, however it has to be considered that very low particle concentrations were measured, about 2 cm[-3]. There were no inter-comparison measurements made for POPS instruments. For more details please see the manuscript discussion."

RC6) *Figure 8, "POPS and OPC-N2 overlap well over eight size bins" may overstate the comparison. The figure is in log scale, and POPS seemed to be a factor of 2 of the OPC-N2.*

AR6) True, it is on average about factor of 2, the expression will be softened in revised version of manuscript.

RC7) *P8, section 3.2.1, The BME280 sensor has a +2 C difference comparing to the MURC. Will this difference constant for the hysteresis temperature profile? Will the observed T and RH data scientifically useful?*

AR7) The temperature difference of 2 C of BME280 when compered to MURC, was the same as factory calibrated Vaisala AQT400 sensor mounted on the same rotorcraft. Many other sensors participated in the comparison showed deviation of same magnitude and direction, for details see Barbieri et al. (2019). The observed difference is very difficult to extrapolate to vertical profile, and most probably it will not be constant in whole column. In general, any comparisons to standards (P, T, RH, winds, particle counts) are of high value. The best way to do it is a comparison with in-situ measurements at different heights on the high towers or against tethered balloon system. During LAPSE-RATE the best we had was MURC with measurements at one height, 18 m AGL.

RC8) *P9, section 3.2.2, it is great to see the new particle formation detection capability developed here. It will be beneficial to correct the over-counting behavior of CPC2 before calculating the delta(CPC). Line 266, if the descending rotorcraft would push aerosol particles downwards. Would the ascending flight push aerosol particles upwards? What is the ascending and descending rate of this flight?*

AR8) The overcounting of CPC2 is given mostly by fluctuations in flow which is not recorded. Both CPCs are intentionally calibrated to different cut-off diameter 7 and 14 nm and if no particles smaller then 14 nm are measured, e.g. monodisperse aerosol is used in lab, the uncertainty in count will be still 10 % only due to flow instability. All CPC data are already corrected since their flow rate deviates from nominal 0.7 lpm. The TSI 3007 is a good instrument, but it was not meant for UAV measurements.

The overall movement of the airmass will be down in any case, 11 kg UAV needs to push 11 kg of air down to keep hovering.

Line 231: "Our ascent rates were approximately 5-8 and 3-5 $m.s^{-1}$ and descent rates were about 2-5 and 2-3 $m.s^{-1}$ for flights with particle module (FMI-PRKL1) and gas module (FMI-PRKL2), respectively.

RC9) *Line 273, the site is about 700 m AGL. Will the 1000 AGL Flexpart dispersion model represent the 700 m AGL condition?*

AR9) Here, our focus is on the air mass origin of the elevated layer of nucleation-mode particles discussed in the previous paragraph. We chose 1000 m AGL to make sure that the Flexpart simulation represents air mass movements for the elevated layer and not the surface layer below 700 m AGL.

In case of such layered structure of the atmosphere as seen on July 18th it is uncertain how accurately the underlying meteorological data can represent the vertical structure of the wind profile. Therefore, running the simulation close to the lower edge of the layer (700 m AGL) would increase the risk of simulating the surface layer instead of the elevated layer.

We have modified line 273 as: "We used the Flexpart dispersion model to investigate the air mass history of the elevated layer of nucleation-mode particles observed on July 18th."

RC10) *P10, section 3.2.3, It sounds that the platform motion has an impact on the POPS data. Please quantify the impactor. Does the author characterize the inlet loss for all the aerosol instruments – CPC, POPS, and OPC? It is critical to know the inlet loss for OPC during the descending and ascending because*

*the concentration is very low – 5 cmˆ-3.*

AR10) Impact on POPS data had only fast movement during horizontal transects with inlet placed horizontally and facing the direction of the movement. There were very few such flights, but we have noticed the bias. This will be clarified in revised manuscript.

The POPS on rotorcraft had horizontally orientated naked inlet facing out front of the rotorcraft. The surface POPS used 45 cm long vertically orientated inlet made of conductive tubing, the penetration through the inlet was estimated ~92% for 3 um particles and better for smaller ones.

Both OPCs, surface and particle module, were used with no additional inlet, those OPCs are not meant to be used with any kind of inlet due to use of fan for aerosol intake. On the rotorcraft the OPC was mounted from the bottom and middle of the carbon plate of the module, thus shielded from airmass movement and propeller eddies.

Each of the CPCs used 30 cm inlet made of conductive tubing, led upwards to the center of the rotorcraft where both lines were merged to additional 10 cm piece of conductive inlet tubing, also facing upwards. Penetration for such inlet is ~90 to 99% for particle between 7-100 nm and ~99 % for 100 nm -1 um.

---

## Author Comment (AC2) · 9 Oct 2020

***Anonymous Referee #2***

*RC1) The manuscript "Measurement Report: Properties of aerosol and gases in the vertical profile during the LAPSE-RATE campaign" by Brus et al. provides a description of UAS platform measurements of particles, trace gases and meteorological parameters in the San Luis Valley of Colorado in July 2018. The measurements were made as part of a multi-institution campaign associated with the ISARRA meeting that was held in Colorado that summer. The authors provide substantial details regarding the instruments, platforms and operation, and as such, the manuscript serves to provide a solid example of the kind of research that can be accomplished, and likely most effectively, using sUAS platforms. The manuscript is well organized and fairly well written, but the readability would benefit from careful copyediting for usage and punctuation. Overall, given the focus of the paper on the description of measurements and measurement systems rather than scientific analysis, I would think the manuscript would be a better fit in AMT than ACP.*

AR1) We thank reviewer #2 for constructive comments, we really appreciate their time spent reading our manuscript. We will use Copernicus free copyediting service if our manuscript proceeds to publication. The submission to a new "ACP Measurement Report" was suggested by Editor as appropriate fit.

*General comments:*

*RC2) The authors should be consistent with the description of the team (FNMI-KSU) or teams (FNMI and KSU), both are used in the manuscript, e.g. L42 "A flight team", L44 "by the FMI and KSU teams".*

AR2) This will be corrected in revised version of manuscript.

*RC3) In terms of lower atmosphere/boundary layer studies, height AGL is more relevant or important to know than absolute altitude (MSL). Similarly, for diurnal processes such as boundary layer development, using local time is clearer than UTC, which requires the reader to calculate the local time to interpret the data.*

AR3) The use of MSL altitude and UTC time was based on LAPSE-RATE organizers and participants decision. The standardization of all data sets produced during LAPSE-RATE and their interpretation was a must. The driving force for such decision were the needs of known in advance end users, the modeling and forecasting community.

We are going to use both in the manuscript text, the AGL and local time in parenthesis.

*RC4) The discussion of the vertical profile observations is mostly descriptive rather than analytical, with the exception of the final paragraph of 3.2.2 (NPF), which would still benefit from substantially stronger arguments to derive the conclusions from the observations. Figures 4 and 5 are somewhat tangential to the focus of the manuscript. The inclusion of the descent data in Figure 9 clutters the picture when it has been argued in the text that the ascent data are believed to be more reliable. A single panel with both ascent and descent for comparison could be included instead to illustrate the description in the text.*

AR4) Yes, we are aware of limited analysis provided in the manuscript, since it was submitted as "Measurement report". We are going to add inter-comparison of particle counters and T, RH, P at surface level just before each flight as suggested by reviewer1.

Figure 4 will be changed to time series, figure 5 will be omitted. Figure 9 will be split to single panels of separate ascent and descent profiles.

*Specific comments:*

*RC5) L14: The opening sentence of the introduction is a bit absolutist. Perhaps "Most air pollution, including both gases and particulates, is released near the surface from various sources." The specific mention of wet deposition seems out of place since it is discussed nowhere in the manuscript, it could be omitted.*

AR5) Manuscript text changed accordingly.

*RC6) L18: The sentence "The atmospheric boundary layer can be investigated. . ." lists many measurement methods and relevant references, but then just stops without any following statement about why this list was presented. UAS are merely included as one item instead of the paragraph being used to set up the paper by describing why one might use sUAS instead of, or to complement, other methods.*

AR6) UAS method will be elaborated in revised version of the manuscript.

*RC7) L46: suggest replacing "leveraging" with "utilizing" or "using"*

AR7) Changed to "using".

*RC8) L72: "employed a parachute system" was the parachute actually used regularly for landing, or was it there in case of emergency (power failure) and not actually needed during the campaign?*

AR8) Parachute system was not used regularly for landing, only in case of emergency to safe costly payload. It will be clarified in revised version of the manuscript.

*RC9) L79: If 15.5 kg GTW / 5.5 kg PW would have been too much for operation at the altitude of the SLV, what were the actual KSU Matrice operational weights for LAPSE-RATE?*

AR9) The actual weight of DJI Matrice 600 Pro was 9.5 or 10 kg depending on battery set used. The payload maximum total weight was about 1.1 kg: DJI Zenmuse X3 (221 g), GoPro Hero 7 (117 g), naked POPS (700 g), and Meteorological sensor (30 g, iMetXQ2, International Met Systems, Grand Rapids, MI, USA) borrowed from Oklahoma State University. Not all payload parts were used together during all flights.

This will be clarified in revised manuscript.

*RC10) L85: "These modules were easily detached. . .swapping between sensor modules but since you were using two aircraft, did you actually swap the payloads? It didn't seem so from the description of operations.*

AR10) True, the modules were not exchanged during the campaign. Will be changed to following:" These modules could be easily detached from the rotorcraft frame, allowing for swapping between sensor modules to meet the requirements of a given flight. However, this was not necessary during LAPSE-RATE since pair of rotorcraft was available".

*RC11) L90: "The CPCs were calibrated" perhaps "were set", "were adjusted"*

AR11) Change to "were adjusted".

*RC12) L92: The voltage applied to the TEC (or Peltier cooler) sets [or regulates], the temperature difference between the saturation and condensation regions.*

AR12) Changed accordingly.

*RC13) L94: given the cut-off (50%?) diameters of 7 and 14 nm how did the temperature instability limit the detection of sub-20 nm particles, how much did the cut-points vary? Was this incorporated in uncertainty, or was the delta-T measured and compensated?*

AR13) The calibration was done the same way as described in Hameri et al. (2002), the uncertainty of D50 values was determined to be +/- 0.8 nm. The minimum and maximum set points for TEC, 1000 and 2000 mV, are the limiting factor to observe full nucleation mode range up to 20 nm. Using lower or higher setpoints would lead to instrument instability, this claim is based on personal communication with TSI technician. Using lower setpoint than 1000 mV could lead in range extension up to 20 nm, but compromising the device stability. We used values in safe range thus assuming the device is operating in steady state stable mode.

This will be clarified in revised version.

*RC14) L96: "The platform was enclosed on all sides except the bottom with polylactic acid (PLA) foam to shield"; "Bosh" → "Bosch"; "pressure (P), temperature (T) and relative humidity (RH).*

AR14) Corrected accordingly.

*RC15) L100: "6 ms and ±1 hPa for P, 1 s and ±0.5 ◦C for T, and 1 s and ±3% for RH."*

AR15) Corrected accordingly.

*RC16) L120: "oxygen"? Assumed to be ~20% of P?*

AR16) Cannot comment on this, it is Vaisala proprietary compensation algorithm, confidential.

*RC17) L128: suggest stating here that the AQT400 was not useful in the SLV environment.*

AR17) This part just describes all sensors, usefulness of AQT400 is clearly discussed in part 4 Concluding remarks line 350.

*RC18) L146: How long was the horizontal inlet?*

AR18) Corrected accordingly to "…naked inlet about 9 cm (3.5 inch) long…".

*RC19) L147: What was the nature/purpose of the POPS "custom electronics"?*

AR19) The term simply refers to POPS being a research instrument and the electronics being custom-made by the manufacturer rather than an off-the-shelf solution.

*RC20) L149: What is the expected effect of the POPS not being shielded from the sun over the duration of a profile? Did you evaluate the effect by running the instrument on the ground for the same period to see if a measurable effect occurred?*

AR20) We expect the effect on the flight POPS to be small due to it being exposed only for short durations and being constantly cooled by an airflow. POPS electronics are rated for up to +85C. The effects on the

ground POPS are harder to quantify. However, given that the ground POPS measurements stayed relatively flat despite the increase in temperature over the duration of the measurement we expect that the effects due to lack of shielding were small.

*RC21) L163: Typically, MSL would be denoted "altitude" and AGL "height".*

AR21) Corrected accordingly: "…achieved altitude was 3201 m MSL (i.e. height 893 m AGL).".

*RC22) L178: "A detailed description of the daily meteorological conditions in the SLV during the LAPSE-RATE campaign"*

AR22) Corrected accordingly.

*RC23) L181: The data presented in figures 4-7 seem to indicate that the surface measurements were not continuous, but only occurred during portions of each day.*

AR23) Correct, "continuous" surface measurements were done with the battery power surface module only during UAV operation. No other surface measurements of aerosols and meteorological parameters were present in place of operation.

*RC24) L219: "according to the US Environmental Protection Agency"; EPA or USEPA are not needed since there is no further reference in the paper.*

AR24) There is a lack of any reference to PM measurements in the SLV, our goal was to provide at least some data. However, the sentence will be omitted in revised version.

*RC25) L230: "based on which the goal" perhaps "the goal of which was to reach the highest altitude possible in a very short time."*

*RC26) L256: From figure 10 it appears that the bias between the two CPCs was fairly systematic, raising a question about either the determination of cut-off diameter or relative sampling efficiency. This potentially affects quantitative analysis of the results, but does not necessarily impact the qualitative conclusions drawn.*

AA26) In Fig. 10c in the boundary layer the difference between the CPCs with 7nm and 14nm cut-offs is less than 10%, which is within the expected accuracy of TSI 3007. Considering that TSI 3007 has higher uncertainty and lower accuracy than a full-size desktop CPC when used in lab, less than 10% of concentration (in left hand side plot), could be considered as an excellent agreement. Compared to delta_CPC > 3km this constant bias is less than 5%.

*RC27) L273: The discussion of the Flexpart back trajectories and interpretation of the resulting footprint (shown in Figure 11) is rather superficial and speculative.*

AR27) Here, our focus is on the air mass origin of the elevated layer of nucleation-mode particles discussed in the previous paragraph. Please see also reply to reviewer1.

Flexpart footprint shows that power plant emissions cannot be ruled out as a source of the elevated layer of nucleation mode particles, which is relevant information with respect to earlier observations discussed in the previous section.

We have modified line 273 as: "We used the Flexpart dispersion model to investigate the air mass history of the elevated layer of nucleation-mode particles observed on July 18th."

*RC28) L293: The description of the effect of aircraft motion on the POPS sampling is somewhat vague. How did the motion cause the effect? Was it the same for the horizontal and vertical inlet configurations (L145-147)?*

AR28) Impact on POPS data had only fast movement during horizontal transects with inlet placed horizontally and facing the direction of the movement. There were very few such flights, but we have noticed the bias. This will be clarified in revised manuscript.

*RC29) L298: "was 5 cm-3, similarly averaged PM1" should either use a ";" or ", while"*

AR29) Corrected accordingly.

*RC30) L304: "evaporation" "transpiration" if just from plants or "evapotranspiration" if from the environment (surface + plants).*

AR30) Corrected accordingly.

*RC31) L310: "from a BME280 sensor that showed a bias of about. . ."*

AR31) Corrected accordingly.

*RC32) Figure 10: Suggest labeling the panels in order (A, B, C) and change the references in the text to match*
AR32) Corrected accordingly.

References:

Hämeri, K., Koponen, I.K., Aalto, P.P. and Kulmala, M: The particle detection efficiency of the TSI-3007 condensation particle counter, J. Aer. Sci., 33, 10, 2002,https://doi.org/10.1016/S0021-8502(02)00090-3

---

## Author Response (AR2)

Answers to Anonymous Reviewer #2.

*I believe the authors have reasonably addressed most of the comments I made to the previous version of the manuscript and think the manuscript is in good shape for publication. I would like to return to two of my (AR#2) previous comments and the author's responses:*

*RC1: The first is RC19 regarding "custom electronics". If the electronics of the POPS instruments was in the form produced and standardly supplied by the manufacturer, then they should not be considered "custom" anymore than the electronics in e.g. the Licor or CPCs. "custom" leads the reader to think the electronics have been modified or replaced so that the instrument is no longer in a standard configuration.*

AA1: The word "custom" was omitted from the sentence starting on line 164.

*RC2: The second is R24 regarding the reference to EPA measurements. I did not mean to suggest that the sentence be removed, but simply that the acronyms following US Environmental Protection Agency were not needed since they were not used subsequently in the paper.*

AA2: The following sentence was added back to line 240: "Continuous air quality measurements are sparse throughout Colorado, and according to The Environmental Protection Agency website an average PM2.5 concentration of $18\pm5$ µg.m$^{-3}$, PM10 concentration $58\pm23$ µg.m$^{-3}$ published as averaged maximum 24-hour concentration."

*RC3: the statement (p 8, line 241-2) modified in response to AR#1 RC6, "overlap within factor of 2" would be more correctly stated "agree within a factor of 2 in the overlapping size range between 0.46 and 3 µm"*

AA3: The statement was changed accordingly on line 243: "The particle number size distributions measured by POPS and OPC-N2 agree within a factor of 2 in the overlapping sizerange between 0.46 to 3.5 µm, and together…"

*RC4: on p 11 (line 343-4) of the revised manuscript, the time is state as "13 p.m. and 12 a.m."--this would be "1 p.m. and 12 p.m.", but might be better noted using 24 hour time "13:00 and 12:00".*

AA4: , Changed accordingly on line 345: "…profiling started earlier in the morning (about 13:00 and 12:00  UTC, 7:00 and 6:00 local time, respectively)…"

*RC5: the authors included material (including plots) in their response to comments from AR#1 that add value to understanding the quality of the measurements and could be useful to include in supplementary information.*

AA5: The supplementary material was created and will be submitted with the new version of the manuscript.

[revised manuscript text omitted]